# Current State of Mugger Populations

**DOI:** 10.3390/ani14050691

**Published:** 2024-02-22

**Authors:** Milena Sylwia Bors, Pogiri Gowri Shankar, Joanna Gruszczyńska

**Affiliations:** 1Department of Animal Breeding and Conservation, Institute of Animal Sciences, Warsaw University of Life Sciences, Ciszewskiego 8, 02-786 Warsaw, Poland; 2Kālinga Foundation, Agumbe Hobli, Guddekere, Shivamogga 577411, India

**Keywords:** conservation, risk assessment, *Crocodylus palustris*

## Abstract

**Simple Summary:**

The mugger is facing complex threats in today’s world. This review aims to assess and characterize the existing world population of muggers in reference to its historical range, as well as perform habitat suitability assessment and threat assessment for each site. Based on extensive literature, we found that the chance of preserving muggers varies greatly among countries and habitats across its range. While two main groups of threats can be defined—habitat degradation and human–crocodile conflict—each site displays different characteristics and severity of these threats. We believe this work can be used as a basis to understand the current state and challenges faced by muggers today.

**Abstract:**

The mugger (*Crocodylus palustris*) is a medium-sized crocodilian inhabiting South Asia. As a result of intensive hunting, its range declined drastically up till the 1970s. Currently, the world mugger population is fragmented and threatened mainly by habitat loss and the consequences of human–crocodile conflict, being classified as Vulnerable by the IUCN. The goal of this paper is to comprehensively determine the mugger’s current range, and assess risks in notable habitats of the species across its range. To determine the range and notable habitats, extensive literature covering surveys, monitoring, population studies and reports of human–crocodile conflict was examined. Habitat suitability and risk assessment were performed by evaluating selected habitats using eight factors: the legal status of the area, elevation, surface water availability, water quality, salinity, availability of nesting and basking sites, interaction with humans and interspecific competition. Based on our findings, the chances of the mugger’s survival varies greatly across its range and the threats they face are complex and often site-specific. Defining these threats is the first step for determining suitable risk mitigation efforts, some of which are explored in this review.

## 1. Introduction

*Crocodylus palustris* is distributed throughout India, Sri Lanka, southern Pakistan, southern Nepal and southeastern Iran [1]. The area of occurrence has continuously declined and the extinction of local populations has occurred during the last few decades, with muggers being extinct in the wild in Bangladesh, Bhutan (in the 1960s) and Myanmar (the last reported sightings took place in 1867 to 1868) (Figure 1) [1]. Population declines have been reported in Sri Lanka, Pakistan and Iran [2,3,4]. In India, there has been a reported rise in numbers from the 1970s due to reintroduction and protection efforts, with the country’s population estimated at 4000+ individuals [5,6,7]. However, *C. palustris* populations are fragmented and isolated within its distribution, mainly due to continuous habitat destruction and fragmentation for agricultural and industrial expansion during the last few decades. There have also been constraints on reintroducing muggers from captive breeding centers in the peripheries of Protected Areas due to opposition from local communities [8]. The observed declines in the numbers, range size and availability of quality habitats has led to the mugger being classified as “Vulnerable” by the IUCN.

Habitat destruction is considered the most important driver of species extinction [9]. Mugger inhabits a vast range of habitats, from freshwater rivers to hyper-saline estuaries in Sri Lanka; however, many of these ecosystems are undergoing degradation. Wetland cover has shrunk by 64% since the year 1900, and 30% of all freshwater ecosystems have been lost since the 1970s [10,11]. These ecosystems are highly vulnerable to global warming, eutrophication, invasive species and pollution. Crocodiles are also semiaquatic, as they thermoregulate and nest on land. Nesting sites need to provide a suitable nesting material, vegetation cover and microclimate, as well as be in proximity of water bodies suitable for hatchlings. The alteration of riverbanks, sand mining, deforestation, noise pollution and free-roaming domestic animals all reduce available nesting sites.

Therefore, the goal of this paper is to determine the status and survival chances of the species by characterizing existing wild and captive populations and assessing threats and habitat suitability, based on which predictions and recommendations can be made.

## 2. Materials and Methods

### 2.1. Literature Review

We conducted an extensive literature review by utilizing several resources. We used the search engine Google Scholar by entering the term *Crocodylus palustris* in July 2023. We then compiled every paper found relating to past and current wild and captive mugger crocodile populations, including reports of human–crocodile conflict. We have also compiled every reference to the mugger in the Crocodile Specialist Group Newsletter from the years 1971 to 2023, as well as every reference to the mugger in CSG Proceedings from 1971 to 2022, available on the CSG official website. Another major resource was composed of reports and management plans shared by the Forest Department and other government bodies, ranging from 1970 to 2023.

The information was then synthesized for each of the six countries within the mugger’s historic range and the information presented following the format co-specifics, historical range and status, current range and status, habitat description and suitability, threats, captive populations, other conservation efforts, prognosis and recommendations.

Historic range and status sections cover information on wild mugger populations before the year 2000, with particular importance to areas where they had disappeared in the wild. Sources published after 2000 are cited if they pertain to range or status before the year 2000 or confirm the disappearance of muggers from an area. Current range and status section covers reports on mugger range and status after the year 2000.

Habitats with mugger presence confirmed after the year 2000 by surveys, other research papers, management plans and reports of attacks on humans, were described and assessed by using the following resources: results of the Google Scholar search engine (by location name), Crocodile Specialist Group Newsletter and CSG Proceedings, working plans and Protected Area management plans provided by the Forest Department, reports and action plans from other government institutions pertaining to pollution, drought and flooding management, reports of international organizations on water safety, as well as news reports on crocodile attacks on humans. All of the resources listed above were then a basis to describe the characteristics and severity of threats, with most the prevalent threats being highlighted.

The captive population section contains information regarding breeding and rearing centers and their stock, while the reintroduction efforts section details the source, quantity, location and timeframes of mugger reintroduction. Other conservation efforts include descriptions of the mugger legal status and both active and passive conservation efforts and strategies.

Finally, after compiling all of the information above, we include a prognosis and recommendation sections for each country, taking into consideration the chance of preserving muggers, major threats and their severity as well as proposing mitigation methods.

### 2.2. Habitat Description and Suitability

Habitat characterization and quality assessment was performed based on the known habitat requirements of the mugger. Since our knowledge of these requirements is based on literature and is incomplete at best, some generalization was necessary. Due to the inability to measure some components of the habitat, as well as clearly define tolerance ranges, each habitat required an individual approach and was assessed according to eight factors listed below.

Status. Based on legislature and management plans and reports, three categories were established—Protected Areas (PAs), Protected Areas under risk of overexploitation and areas not protected by law. Distinguishing between Protected Areas and Protected Areas under risk of overexploitation was considered necessary, as 45% of the PAs mentioned house a considerable population of tribal people and/or are in the vicinity of large congregations of humans living in poverty that rely on PAs for resources such as timber, firewood, plants, game and fish. While local tribes have been historically sustainably using these resources and usually have a legal right to do so, people living in poverty in and around PAs acquire them illegally with a destructive effect on the biodiversity. It is highly unlikely that with the rising human population in all discussed countries, this practice will cease; therefore, a distinction was made between two types of Protected Areas. Assigning PAs to one of the two categories was based on management plans, reports and research papers.

Elevation. Based on IUCN reports, the mugger was previously believed not to occur higher than 420 m above sea level (masl) [12]. Muggers have since been observed on higher elevations, such as at 822 masl in Similipal Tiger Reserve in Odisha, India. The misconception that muggers do not occur higher than 420 masl may be caused by fewer suitable water bodies at higher elevations, as crocodiles do not inhabit streams with strong currents.

Surface water. The quantity of surface water is another important aspect. Muggers are known to inhabit various types of water bodies; therefore, rivers, lakes, ponds, man-made reservoirs, lagoons and estuaries are all considered suitable, excluding streams and rivers with strong currents. Strong current is avoided by crocodiles, due to the higher energy required to swim. The stability of these water bodies is taken into consideration, although seasonal droughts do not exclude habitats, as long as some water bodies in the vicinity maintain water. The preferred depth varies between different reports, but an average of 2 to 4 m seems to be favored [13].

Water quality. Water quality is determined by reports of contamination with sewage discharge, heavy metals, plastic, industrial and household waste. As apex predators, crocodiles are susceptible to bioaccumulation of heavy metals [14]. While they are less susceptible to bacterial infections thanks to the antibacterial activity of crocodilian plasma, high concentrations of bacteria can still result in infections presenting with symptoms [15]. Bacteria such as *Salmonella* spp. and *Escherichia coli* can cause infections resulting in death, to which hatchlings are particularly susceptible [16]. Microbiota in the soil can also affect the viability of relevant traits in snakes; therefore, it is likely this can also occur in crocodiles [17]. Crocodiles can also swallow various types of trash, such as plastic bags, that inevitably lead to the death of an individual. Sedimentation is also taken into consideration, as it causes water levels to rise, flooding previously viable nesting sites [18]. The water quality is related to seasons, as a lower flushing time results in the retention of pollutants.

Salinity. Muggers prefer freshwater habitats, although they do occur in hyper-saline estuaries in Sri Lanka [19]. Therefore, salinity is of some importance, although high salinity does not exclude a habitat from being suitable for *C. palustris.* Salinity tolerance likely differs among populations, as muggers are known to inhabit saline habitats in Sri Lanka, but not in Iran; therefore, changes in salinity due to rising sea levels and pollution must be also considered.

The availability of nesting and basking sites. While studies have attempted to characterize basking and nesting site selection by muggers, the results are impaired by large numbers of possible habitat factors as well as population-specific preferences [18]. It is clear that muggers tolerate and possibly even prefer steeper inclines than other sympatric crocodilians, most likely due to their high walk, compared to crawling species [20]. They bask on a variety of substrates, including sandbars, fallen logs, clay and sand. Choudhary and Choudhary [20] noted their preference for river banks, as well as a slight seasonal change in preferences for basking sites. They also noted a preference for sand and clay for digging nests. Muggers in Iran prefer a mean vegetation cover of 35% and a mean slope of 25–35% [13]. Due to unclear habitat preferences, the nesting and basking site availability will be mostly determined by human disturbance and flooding.

Interaction with humans. Muggers can adapt well to certain anthropogenic and even urban habitats, as established by the presence of a large population in the city of Vadodara, Gujarat. It is, however, a suboptimal habitat, since sharing space with humans leads to the exacerbation of human–crocodile conflict. The public opinion on crocodiles is an important factor in implementing and executing any conservation program, as crocodiles are killed in retaliation or as a preventative measure. Human sympathy could also contribute to alleviating some of the risks that crocodiles face, such as habitat loss and degradation and contamination. Even in areas with a lower density of human population, muggers are at risk of drowning in fishing nets, which is considered to be one of the biggest threats to the species [1].

Interspecific competition. The occurrence of other species of animals occupying similar ecological niches was considered a factor due to the adverse effects it could have on *C. palustris*, such as saltwater crocodile (*Crocodylus porosus*). The opposite situation was also taken into account, as muggers can compete with other endangered species, such as gharial and big cats [1]. While populations of *C. palustris* sharing habitats with the gharial should still be preserved, they should not be restocked, as there are reports that muggers supersede the gharial [20]. Areas established to protect other species of endangered fauna sharing a niche with muggers, such as river dolphins, should not be considered for reintroduction.

## 3. Country Summaries

### 3.1. Sri Lanka

The mugger is one of the two crocodile species in Sri Lanka, the other being the saltwater crocodile (*Crocodylus porosus)*.

#### 3.1.1. Historical Range and Status

Sources from the 19th century and the beginning of the 20th century state that the mugger was widely distributed in the dry zone of Sri Lanka and could be found in virtually every freshwater body. However, due to indiscriminate hunting for skins and meat, their numbers have declined, with researchers noting an obvious decline in numbers in the middle of the 20th century [21,22]. In 1930, Deraniyagala wrote “the species which was so common in 1925 is now rarely found in any numbers and specimens 3 m long are very scarce”, pointing to a large population decline in just five years [23]. Between 1920 and 1940, mugger numbers declined with the increased export of raw skins [24,25,26]. *Crocodylus palustris* previously inhabited the Jaffna peninsula, where it no longer occurs. Comprehensive surveys carried out by Santiapillai et al. suggest muggers may also have gone extinct locally in the Kelani Ganga, Polwatta Oya, Karambalan Oya and Maha Oya river systems [2]. Surveys carried out by Whitaker and Whitaker in 1997 to estimate its total population in Sri Lanka suggested the total population was as high as 3000 individuals [25,27].

#### 3.1.2. Current Range and Status

At present, the mugger is mostly confined to the first peneplain of Sri Lanka (Figure 2) in the dry zone below 100 masl [28]. It inhabits freshwater streams, rivers, canals, tanks and even hyper-saline estuaries [22]. There appears to be a 60% decline in muggers during the past 20 years in the country [28]. Madawala et al. reported the presence of muggers in all 10 administrative districts of the island [29]. Sentapillai and de Silva have reported sightings from 104 locations [30]. The most important regions in crocodile conservation in Sri Lanka are Protected Areas, particularly the Wilpattu National Park and Yala Protected Area Complex. During the previous study, 500–600 adult and sub-adult muggers inhabiting Block One of the Yala National Park were observed [30]. Over 100 muggers were recorded in the Katagamuwa tank alone. This tank is situated close to Block One of Ruhuna National Park [28].

#### 3.1.3. Habitat Description and Suitability

Sri Lanka, except for its high hill areas, has a year-round tropical climate. The dry zone comprises the coastal and low country area, located in the island’s north, north–central and eastern regions [31]. The wet zone covers approximately 20% of the land in the southern, western and central parts of Sri Lanka.

There are 103 natural river basins in Sri Lanka, with a total length of about 4500 km [32]. They belong to 34 major river drainages, 7 in the dry zone, 2 in the dry and wet zones and 25 in the wet zone. The wet zone rivers are perennial, and the dry zone rivers shrink in the dry season [25].

There are no natural lakes on the island. There are, however, around 12,000 water basins of anthropogenic origin. Tanks, which were constructed between the 5th century BC and the 14th century AD provide greatly expanded habitats for muggers, becoming their primary habitat in Sri Lanka [25,26].

Although it is difficult to assess the water quality of natural reservoirs, due to a lack of monitoring data, concerns have been raised over nitrate and bacteria contamination resulting from poor sanitation [32]. There is also a concern for toxic chemicals from the industrial and agricultural use of land as well as the eutrophication of lakes [32].

Habitat suitability assessment in Sri Lanka is presented in Table 1.

#### 3.1.4. Threats

While *C. palustris* is classified as vulnerable (VU) by IUCN, according to Santiapillai et al., within Sri Lanka, the mugger meets IUCN criteria for being endangered (EN) [28].

The drastic decline in mugger numbers has been attributed to the high demand for their skin and meat. According to Whitaker and Whitaker, fishermen in Sri Lanka used to kill crocodiles for meat in the 1980s, sometimes as many as 20 in a day [38].

Another threat to crocodiles is caused by the proximity of their habitats to human settlements. The island has an ever-growing human population, with the World Bank reporting a rise from 10 to almost 22 million in the years 1960–2022. This led to the exacerbation of human–crocodile conflict, particularly around water bodies. Madawala et al. identified five regions with particularly high human–crocodile conflict: Weerawila-Hambantota-Tissamaharamaya, Kumana-Panama-Lahugala, Bibila-NilgalaAmpara, Udawalawa-ThanamalwilaLunugamvehera and Anuradhapura-Kurunegala-Dambulla [29]. During a five-year study over the period of 2008–2013, carried out by Madawala et al., 165 cases of intentional killings of muggers were reported [29]. Approximately 29% of these killings were attributed to preventing future attacks on humans, of which 26 took place during the study period. Additionally, 36 attacks on pets and livestock were recorded. This means that according to these data, more crocodiles were killed to prevent attacks than the number of actual attacks on humans.

Another outcome of the growing population is habitat degradation, through converting land for agricultural and industrial purposes, as well as the draining of wetlands. This creates a cycle in the result of which crocodiles may move even closer to human settlements, since they rely on the proximity of water bodies, further inflaming the human–crocodile conflict. The sizes of adult crocodiles, together with their territorial nature, necessitate relatively large and diverse areas of undisturbed wetlands for the maintenance of large populations [39].

Intensive inland fishing also contributes to decline, through a reduction in food availability and more importantly, the accidental drowning of crocodiles in fishnets, especially gillnets. Fishermen may also deliberately kill large numbers of crocodiles, believing they compete for fish [26,29].

As with many other species, the mugger is also affected by climate change. While concerns have been raised over the effects of climate change on the sex determination of offspring, it is still too early to assess that impact. Climate change is also contributing to an increased risk of drought. The increased variability of rainfall has been linked to extreme surface runoff, contributing to the aggravation of droughts. In the period from May to August (Yala season) the entire dry zone is at risk of droughts [40]. While short mild droughts may be beneficial for crocodiles, with saline lagoons and lakes increasing in salinity, therefore causing mass deaths in fish, and providing a considerable food source, long-lasting droughts result in habitat loss, food shortage and dehydration.

#### 3.1.5. Captive Populations

Considering the demand for crocodile skin and meat, one might assume that captive breeding could be a remedy allowing for limiting poaching. Unfortunately, breeding and utilizing wildlife in captivity is unethical in Sri Lanka’s Buddhist culture. It is unlikely that a considerable captive population will be established in the near future.

#### 3.1.6. Other Conservation Efforts

Crocodiles became legally protected in Sri Lanka in 1938, but law enforcement is ineffective if it is against public opinion. Muggers in Sri Lanka have been protected by the Fauna and Flora Protection Ordinance (Act no: 44 of 1964) since 1964 in response to population decline due to indiscriminate hunting; however, they can still be hunted by license holders [41]. The export of crocodile skins has been banned. Nevertheless, given that reservoirs inhabited by crocodiles outside the Protected Areas come under the Irrigation Department and not the Department of Wildlife Conservation, crocodiles are vulnerable to poaching.

Sri Lanka has made considerable progress in establishing Protected Areas which cover over 12% of the land area, but the system of Protected Areas is not comprehensive as far as crocodiles are concerned and remains inadequate in western, southwestern and southern areas.

While Act 1 of the Customs Gazette of 1969 prohibits the export of crocodile skins, Sri Lanka has lately become a transit point in the global trafficking of endangered species because of the rather poor enforcement of laws.

#### 3.1.7. Prognosis

Previous population surveys in 1977 estimated the number of crocodiles to be about 3000 individuals, with subsequent survey results published in 2000 estimating that number to decline by 60%, reaching a minimum of 1220. With human population growth further escalating in the last two decades, resulting in further habitat degradation, this number is likely much lower today. While populations in the two mentioned national parks have a high chance of survival, being less threatened by habitat loss, crocodiles inhabiting unprotected areas are considerably more at risk. There seem to be no migration corridors between the national parks, which likely contributes to lower genetic diversity. It would be possible to rehabilitate these populations, considering the number of available habitats in the form of man-made tanks. However, it would first require intense education efforts to teach local communities safety measures to coexist with crocodiles and the role of crocodiles in maintaining stable ecosystems.

### 3.2. Bangladesh

Now considered most likely extinct in the wild, muggers historically have inhabited Bangladesh, along with its sister species, the saltwater crocodile.

#### 3.2.1. Historical Range and Status

According to records, muggers used to be abundant in Bangladesh [42]. Unfortunately, due to the species’ extinction in the 20th century, little to no records remain of the mugger’s historical range in Bangladesh. Some experts believe muggers inhabited the Sundarbans fringes; however, this remains speculative [43]. The mugger was declared effectively extinct in Bangladesh in the 20th century [44,45]. An unsuccessful breeding program resulted in three individuals being held in semi-captivity at Khan Jahan Ali Majar Pond, Bagerhat [40].

#### 3.2.2. Current Range and Status

In 2005, 29 individuals hailing from the Madras Crocodile Gene Bank, Tamil Nadu, India, were released in Bangabandhu Sheikh Mujib Safari Park [46].

#### 3.2.3. Habitat Description and Suitability

Bangladesh is a low-lying riverine country, located in the northeastern part of the South Asian subcontinent. Bangladesh comprises 14.4 million hectares of land, with three physiographic regions: flood plains, terraces and hills, with flood plains covering 80% of the country [47]. The climate is subtropical, with the average temperature ranging from 10 °C in winter to 38 °C in summer [48]. Most rainfall falls during the monsoon season (June to September), on which the water supply is dependent. Due to massive amounts of rainfall falling only during these four months, the country battles with both floods and droughts. The most prevalent ecoregion in the country is the lower Gangetic Plains with its moist deciduous forests (77.1% of land cover), followed by the Sundarbans mangroves, the largest mangrove forest in the world (11.3%). Two other ecosystems of notice are Sundarbans freshwater swamp forests (5.5%) and the Mizoram–Manipur–Kachin rain forests (5%). Bangladesh lies on floodplains of the three major rivers—Ganges, Brahmaputra and Meghana, as well as over 430 smaller rivers. Due to erosion and accretion, the country is losing its flood plains, with a total loss of over 88,000 hectares between 1973 and 2014 [48].

#### 3.2.4. Threats

Since the mugger is considered to be extinct in the wild, we will consider threats and contraindications to the reintroduction of mugger. The first major factor preventing reintroduction is the fact that breeding programs in Bangladesh have been unsuccessful, possible reasons for which are presented below. The second is the lack of suitable habitats, with most Protected Areas in Bangladesh threatened by overexploitation and human encroachment. There is also a high risk of developing human–crocodile conflict due to human population density, as observed with a declining population of the saltwater crocodile in Bangladesh, most likely caused by indiscriminate killings [49].

#### 3.2.5. Captive Populations

Current information on captive stocks is unknown; however, in 2009, zoos in Bangladesh were housing 40 adult and 28 hatchling muggers [1]. Two habituated individuals remain at Khan Jahan Ali Majar Pond.

#### 3.2.6. Reintroduction Efforts

As stated above, there have been attempts to reintroduce muggers in Bangladesh, first at Khan Jahan Ali Majar Pond, where three individuals kept in semi-captivity did not reproduce successfully. The reintroduction of a larger number of crocodiles (29 individuals) took place in 2005 in Bangabandhu Sheikh Mujib Safari Park. Out of those individuals, 24 were females and five were males [42]. Although the reintroduced crocodiles attempted to reproduce, laying 200–300 eggs each year, they produced no hatchlings. After the first reproduction attempt in 2006, at the end of their usual incubation period, the eggs were found to be rotten, as was the case in the following years. Two separate studies attempted to determine the factors leading to breeding failure. Their findings upon scrutiny were found to be inconclusive and even contradictory. Masum et al. deemed the habitat sufficient for nesting in regard to space, the type of terrain and shading; however, Hossen et al. stated that the volume of the water body and particularly its depth was insufficient for population density in the semi-captive enclosure [42,50]. Citing Joanen and Mcnease and King and Dobbs, Hossen et al. stated that the water should be at least 0.9 m deep, while it was reported to be too shallow in the entire enclosure; however, the authors of both of these papers based their recommendations for enclosures on species other than muggers [50]. Hossen et al. also found little to no nesting material in the nests, pointing to the lack of suitable nesting material in the enclosure, which would leave females unable to control the nest’s temperature [50]. The most important reasons suggested for the inability to reproduce were temperature and humidity. The standard incubation temperature for mugger ranges between 28 °C and 34 °C. Both Masum et al. and Hossen et al. reported that the field temperature in Bangabandhu Sheikh Mujib Safari Park exceeded that range during the incubation period [42,50,51,52,53]. While muggers as a hole-nesting species can manipulate the temperature to a certain extent via the depth, type of nesting material and shading, it appears their efforts were insufficient. Both studies also noted that the field humidity for egg incubation was lower than the desired 90–95%, although it is important to note that field humidity and soil humidity may vary greatly [51,52]. Hossen et al. also noted heavy rainfall during the incubation period that could adversely affect hatchability [50].

It is important to note that these crocodiles came from Madras Crocodile Gene Bank in Tamil Nadu, India. While it is a good practice to reintroduce individuals coming from populations genetically similar to ones that have gone extinct in the region, that was not possible in this situation due to the extinction of Bangladeshi muggers in the wild. The incubation temperature range cited above was determined by studies performed on muggers in Crocodile Gene Bank, where Bangladesh’s captive crocodiles hail from. It is possible that muggers hailing from Tamil Nadu, where the daily mean temperature is 34.55 °C on average in April, they did not adapt sufficiently to the climate in the safari park, where the field temperature often exceeded 35 °C during the incubation period.

#### 3.2.7. Other Conservation Efforts

Due to being extinct in the wild, establishing a successful breeding program is currently the only possibility of conserving the species in Bangladesh.

#### 3.2.8. Prognosis

While identifying habitats viable for mugger reintroductions may prove to be a challenge in itself, no such efforts can be taken until the captive crocodiles breed successfully in Bangladesh. That requires further studies on the causes of failure to reproduce in Bangabandhu Sheikh Mujib Safari Park. In addition to on-site studies on reproductive behavior and nesting ecology, we suggest genetic studies to ascertain whether wild populations originating from the northern parts of the range, such as the Indian states of Uttarakhand or Uttar Pradesh, might be more viable to establish a breeding program in Bangladesh.

### 3.3. Iran

Iran represents the western-most range of mugger, the only crocodilian species in the country.

#### 3.3.1. Historical Range and Status

It seems that the mugger’s range has not declined considerably in Iran, as no historical records of the species outside of the currently inhabited Sistan–Baluchestan province (southeastern Iran) were found. The earliest record on the number of crocodiles in Iran estimated their population to be 50 individuals [53]. A later study estimated 50–100 individuals [54]. During the survey carried out in 1992, 118 muggers were observed [55]. They were most abundant in the Sarbaz and Kaju rivers and their surrounding ponds. In 1999, the results of another survey, encompassing largely the same area, yielded a total population estimation of 200–300 individuals [56].

#### 3.3.2. Current Range and Status

The mugger range in Iran is limited to the Sistan–Baluchestan province (Figure 3) [57]. All surveys on Iranian muggers conclude three main rivers of Sistan–Baluchestan province, Sarbaz, Kaju and Bahukalat, and their related headwaters and ponds, to be the main habitats of this species in Iran. Newer surveys have found other populations along the Nahang river and nearby water bodies [58]. According to Mobaraki, the main distributional area starts near Rask and ends around Kollany village [58]. The most recent survey reported a total of 326 crocodiles sighted in rivers, reservoirs and ponds. The highest counts were reported from Phishin Dam Reservoir (over one-third of the total number) and Shirgovaz Regulatory Dam Reservoir. This survey led to the estimation of the total population to be about 500 nonhatchlings. This rise from the 2002 survey results suggests an increase in population size, yet it should be noted that the previous survey took place after six years of prolonged drought [3,56]. Mobaraki notes that Iranian muggers are divided into several scattered sub-populations, due to dam constructions [58]. There are no records of Iranian muggers inhabiting saltwater habitats [58]. Such behavior may occur in areas near the Iranian–Pakistani border, although due to the inaccessibility of this border, carrying out surveys in the region is difficult due to restrictions.

#### 3.3.3. Habitat Description and Suitability

Iran is an Asian country covering over 1.6 million km^2^. It lies on the world’s arid belt and comprises mostly rangelands (52%), deserts (20%) and forests (9%) [59]. Protected areas cover 10% of the country [59]. Its average rainfall is 250 mm (less than one-third of the world average) [59].

The majority of muggers’ range in Iran lies within the Gando Protected Region (465,000 ha), established in 1971 primarily to protect muggers. Part of the region (75,000 ha) was also designated as the Govater Bay and Hur-e Bahu Ramsar site in 1999. The Gandou Protected Region comprises mostly the arid mountain and semi-arid desert habitats straddling the three major rivers. Wetlands occur along these rivers and estuaries, including freshwater pools and marshes, mangrove swamps and intertidal mudflats [60]. Elevation varies from 0 to 50 masl and occasional isolated small mountains are found [61].

Crocodiles live along the three major rivers—Kaju, Sarbaz and Bahukalat. Three major water basins exist in the Gandou Protected Region boundaries—the Persian Gulf, Eastern Border basin and Central basin [62]. Almost half of the country’s renewable water resources come from the Persian Gulf. The upstream Sarbaz River has a running bed and few ponds; therefore, only a small population of crocodiles live there [58].

Habitat suitability assessment in Iran is presented in Table 2.

#### 3.3.4. Threats

The Iranian population of muggers is at the extreme western range of the species, making it somewhat unique. At the national level, muggers may meet the criteria for species at high risk, and therefore be nationally critically endangered [66].

The biggest threats to muggers in Iran are droughts, floods, climate change and the pollution of aquatic ecosystems.

In the last century, the human population of Iran has increased about six-fold [67]. Such rapid growth puts an obvious strain on the country’s natural resources, resulting in water supply depletion and converting natural habitats to agricultural land. Interestingly, even though crocodiles live in close vicinity to human settlements and sometimes even within their boundaries, killing and poaching is not much of an issue in Iran. As a result of cultural beliefs, crocodiles are respected and people refrain from harming them. Many ancient cultures relied on crocodiles to lead them to water and were aware of their positive influence on freshwater habitats. Reports of crocodile attacks in the regions are extremely scarce, with only two known casualties [58,68]. The biggest factor in human–crocodile conflict are attacks on livestock. People do not retaliate by killing crocodiles; they do, however, build obstacles to prevent crocodiles from reaching the livestock. News reports are suggesting a rise in the frequency of attacks on humans, due to drought pushing crocodiles further into human settlements; however, as of this writing, this is not supported by research.

Water availability in Iran is an issue due to difficulty in controlling water withdrawal and pollution sources, the inadequate monitoring of water bodies and inadequate guidelines and standards [67]. Further worsened by climate change and periodic droughts, Iran is under considerable water stress. Rising temperature leads to higher evaporation, which continuously depletes Iran’s water resources [69]. The country’s water-related issues include desiccating lakes, diminishing rivers and declining groundwater resources. The biggest culprit of water withdrawal is the agricultural sector, which is only increasing along with the country’s rising population. According to the United Nations water scarcity index, Iran’s water resources are in critical condition. The NRC pointed out a 42% reduction in the river flow in the past 20 years [68].

Drought is deadly mostly to hatchlings and juveniles, but adult crocodiles are also affected. Small ponds along the major rivers dry very rapidly, forcing crocodiles to migrate and gather in large reservoirs of big dams. Since the climate forces crocodiles to migrate, muggers become victims of traffic [70].

Floods are not as common as droughts and take short episodes, but the flooding of riverbanks during nesting season can cause huge damage. Occasionally, floods carry crocodiles from Phishin Dam downstream, where they become entrapped in an overflow tank, and would die of starvation without human intervention [71].

Apart from water scarcity, its quality is also an issue, with heavy metal contamination in known crocodile habitats [65]. As aquatic apex predators with a long lifespan, crocodiles are highly susceptible to bioaccumulation.

Their population is fragmented between drainages and dams; therefore, gene flow is impaired, which may lead to inbreeding depression in the Iranian muggers.

#### 3.3.5. Captive Populations

Dargas Centre, Rikokash Centre and Gando (Negour) Centre were established for rehabilitation, keeping and rearing juvenile crocodiles. In 2013, they were hosting 50 crocodiles [58]. As of 2018, the Rikokash Centre itself hosts 70 crocodiles [72]. Some of the captive crocodiles have laid eggs, although their hatchability remains an issue. Although one female in Rikokash Centre managed to successfully reproduce for four consecutive years, it failed to produce offspring in the next years, along with other females. The most likely cause of failure to hatch was overheating [58]. These centers also serve an educational role, raising public awareness and engaging the local community in conservation efforts. Currently, there are no plans to reintroduce captive crocodiles in Iran.

#### 3.3.6. Other Conservation Efforts

The mugger is listed as an “Endangered species” and protected by the law in Iran. The killing and capturing of crocodiles has been an offence punished by a 100 million RI fine since 2013 [58]. There are attempts to soften human–crocodile conflict with a compensation program by the Iranian Environment Department for livestock losses caused by crocodiles [73]. Troublesome animals are also relocated, especially during droughts and temporarily held in rehabilitation centers or relocated to habitats distant from human settlements. A management plan has been submitted to the Environment Department [3]. Planned activities consist of research, the conservation of the population in the natural habitats, captive breeding programs and public awareness and ecotourism.

#### 3.3.7. Prognosis

While surveys suggest a rise in the wild mugger population in Iran, it may be only a temporary phenomenon. Mobaraki suggested the rise may be due to the fact that the previous surveys took place during prolonged drought [57]. It is therefore likely that mugger numbers dwindle during droughts. Considering the bleak condition of water resources in Iran, this unique population may be very well at risk of extinction in the not-too-distant future. Water security should be a priority to conserve the wild population of crocodiles and their habitats; however, this issue requires an interdisciplinary approach and vast changes to employed policies. The expansion of current rehabilitation centers to host a larger number of crocodiles and ensuring their ability to reproduce in the centers should be one of the priorities to ensure the survival of the genetic pool of the westernmost mugger population.

### 3.4. Pakistan

The mugger is the only representative of the order Crocodilia in Pakistan, the other historically occurring crocodilian species, the gharial, being extinct in the wild.

#### 3.4.1. Historical Range and Status

In 1980, muggers were reported in Baluchistan province in the rivers Nari, Hub, Fitiani, Hingol, Dasht, Nihang and Kuch Kuar [74,75].

In Sindh province, wild populations were recorded in Chotiari Wetland Complex, Deh Akro II Nawabash and Nara Desert Wildlife Sanctuary [76,77].

Muggers were plentiful in Sindh in the 1930s [78]. According to the Crocodile Specialist Group, in 1972, crocodiles in Pakistan were on a significant decline “resulting in sparse scattered populations and a few stray animals” [74]. A survey from 1985 counted 71 crocodiles in Sindh and Baluchistan provinces and estimated the total population to be over 185 individuals [74]. A survey taking place in 1988–1990 focused on Deh Akro 2 Nawabshah recorded about 2000 individuals from this area alone.

A somewhat unique population inhabits a pool at the Manghopir Karachi Mango Pir shrine. This population is very old, as suggested by archeological discoveries of crocodile worship, dated as far back as 2500–1700 BC [76]. In the 1940s, that population counted 40 crocodiles, but in the 1960s, it dropped to just three individuals [76].

#### 3.4.2. Current Range and Status

The mugger in Pakistan inhabits mostly the lower Indus Valley (Figure 4). The IUCN Red List assessment states that the total population is estimated to be around 600 individuals as of 2013.

Small populations are present in the rivers Hub, Hingol, Fitiani, Basol and Dasht [66]. To our knowledge, most of these rivers were not recently surveyed. In 2007–2008, surveys counted 99 crocodiles from 21 ponds located along Dasht River [4]. Hingol River was subject to severe drought in the years 1999–2004, resulting in crocodiles out of water being hunted for skin [76].

In Deh Akro II Wildlife Sanctuary, 189 specimens were counted [79]. Chang et al. state that crocodiles in the Nara Canal, Deh Akro 2 and Chotiari Reservoir were present in over a thousand as recently as early 2000 [79]. The 2012 survey confirmed the presence of crocodiles in 17 lakes. Out of those, 53 crocodiles were recorded from Wasoo Lake, making it the most densely populated water body in the whole sanctuary. Another important location is Chach Lake, where the highest number of hatchlings was recorded, suggesting this lake is an important breeding habitat [79].

Another stronghold of muggers in Pakistan is the Nara Desert Wildlife Sanctuary. Chang et al. counted a total of 326 crocodiles from 19 lakes and wetlands in the sanctuary [80]. Over one-third of the crocodiles were found in Pirana Pitan.

In the Chotiari Wetland Complex, surveys conducted in the years 2006–2009 counted a total of 66 individuals from 14 lakes and wetlands [81]. During this time, a survey of Haleji Lake Wildlife Sanctuary was also carried out. The survey counted 269 crocodiles and Chang et al. reported this as a rise in population due to successful management [81].

Recently, a new location was found in the Nari Gage canal, including nests, suggesting muggers reproduce in the area [82]. Mobaraki et al. list Mehrano Wildlife Sanctuary as another location, though no other records were found [65].

The Mango Pir shrine crocodiles living in an isolated freshwater lake can be considered a semi-wild population since crocodiles are believed to migrate to the lake naturally.

#### 3.4.3. Habitat Description and Suitability

Pakistan is the second-largest South Asian country. The climate is continental, with an extreme temperature amplitude throughout the year [83]. Due to variations in rainfall between the dry and monsoon seasons, the country experiences periods of both flooding and drought.

The country comprises three ecological zones, namely mountainous ranges, foothills and the Indus plains. Due to mugger’s habitat requirements, only the Indus plains are inhabited by crocodiles. Habitat types in the Indus plains include riverine tracts, seasonal inundation zones and swamps, tropical thorn forests and sand dune deserts [83].

The main source of surface water is the Western Himalayas, carrying water through the Indus Valley [83].

Habitat suitability assessment in Pakistan is presented in Table 3.

#### 3.4.4. Threats

Pakistan is experiencing more frequent and prolonged droughts due to climate change. According to the Emergency Plan of Action, historically, droughts have appeared in a 16–20-year cycle, while in 2018 alone, three drought alarms were issued by the Pakistan Meteorological Department [92]. The regions most affected are the Sindh and Balochistan provinces. While Pakistan’s government with international help works to alleviate the problem, it is unlikely that the issue will disappear in the future. Seasonal flooding also occurs, threatening nests.

Pakistan’s steadily growing human population results in a higher demand for water and agricultural land, both contributing to habitat degradation. For instance, in 1988, Deh Akro II Desert Wildlife Sanctuary consisted of 45 wetlands and water basins, but in 2021, only 32 of these remain. Kunbhar made a bleak prediction of muggers going extinct in Pakistan in 8–10 years due to the lack of suitable habitats [93].

Poaching and killing to protect livestock is a major issue in preserving crocodiles in Pakistan. It is believed that this was the reason for the drastic decrease in the mugger population in the past century [78]. There are reports of juvenile muggers smuggled from Pakistan to Iran [4]. Killing crocodiles is illegal in the country, but there are reports of the inadequate enforcement of the law, particularly a lack of rangers and patrols necessary [91]. When the Dasht River dries, crocodiles are left with little choice but to prey on livestock [4]. Fishermen kill crocodiles, either passively by setting fishnets or purposely, perceiving them as competition. Crocodiles are also killed for no gain other than for sport. It seems that poaching has reduced significantly in Sindh since the implementation of the preservation program in 2006–2009, but these efforts failed to bring a significant change in Baluchistan [66]. It is difficult to assess the frequency and severity of crocodile attacks on humans, due to many of them being unreported and contradictory claims from literature. Sideleau reported only two attacks from 2011 to 2021, both fatal [94]. Conversely, CSG reports that in 2006, crocodiles attacked nine people at Haleji Lake, killing one, and in 2020–2021, four people were attacked at the Nara Canal; two were killed [66]. Mobaraki et al. states there were 16 reports of attacks from 2011 to 2021 [66].

#### 3.4.5. Captive Populations

There are seven centers and farms rearing and breeding crocodiles in Pakistan. Chang reports that Khar Center, Haleji Centre and New Jatoi Farm are overstocked, with no clear reintroduction plan in the near future [85]. Crocodiles are also kept in Samzu Park and Khar Center. We were not able to obtain reliable records about stock in Joccyanwala and Gatwala, located in Punjab [95]. There is no single agency responsible for tracking the success of restocking animals, which makes reintroduction difficult to plan and the success rate unclear. There have been efforts since 1987 to obtain crocodiles from Madras Crocodile Gene Bank in Tamil Nadu, India. In 1997, the Punjab Wildlife Department purchased 300 crocodiles from the Government of India, but apparently they were not delivered [95].

Khar Center in Khirthar National Park is located in the southwestern region of Sindh. In 2009, the center housed 40 crocodiles, of which 30 were adults [91].

New Jatoi Farm is a non-profit private farm, operating since before the 1970s and established by a local landlord [85]. This farm consists of three ponds, where adults, juveniles and hatchlings are kept, respectively. In 2015, the farm’s stock consisted of 45 adults, 44 juveniles and 76 hatchlings [85]. While crocodiles seem to have no issue with breeding, only a small fraction of eggs is collected to hatch, due to overstocking.

Karachi Zoological Garden was founded in 1878. A reptilian house was established in the garden in 1992 [91]. In 2009, the center housed 39 crocodiles, of which 23 were adults [91].

Samzu Park housed 14 crocodiles in 2009 and eight of them were adults [80].

#### 3.4.6. Reintroduction Efforts

In 1983, the government of Pakistan approved a five-year project to rear and release crocodiles into the wild, although no records of the number of animals reintroduced or of the success of the reintroduction were available [96]. There are mentions from 1987 of muggers introduced in Haleji Lake [97]. Khan reports that crocodiles have established themselves in the lake, supported by the existing population [97].

#### 3.4.7. Other Conservation Efforts

Several actions are being taken to protect muggers in Pakistan. Muggers were granted legal protection in the 1960s. As mentioned above, that did not deter people from hunting crocodiles, although, as a result, poaching in 2006–2009 in Sindh province decreased. Captive centers function well and crocodiles reproduce successfully, but there are no clear reintroduction plans or institutions responsible for keeping track of the survivability of the released crocodiles. The government protects crocodiles indirectly by preserving their habitats, through implementing conservation programs such as the Pakistan Wetlands Programme and creating and implementing strategies for appropriate water management and water quality control.

#### 3.4.8. Prognosis

The situation for muggers does not look promising in Pakistan. All threats to crocodiles, that is, poaching, habitat degradation and drought risk, are severe and create a deadly combination for muggers. To preserve crocodiles, maintaining their current habitat is a priority, although it seems current conservation efforts may only delay the inevitable. One thing going for them is that muggers in Pakistan inhabit mostly Protected Areas, which are already under legal protection and are often national heritage, thus there is high interest in preserving them. That interest alone is, unfortunately, not enough to prevent habitat degradation through conversion into agricultural land, water shortage and insufficient water quality management. Overstocked captive farms and centers should be used and crocodiles released into the wild in areas with a lower risk of inflaming human–crocodile conflict. These animals should be monitored to assess the success rate. Extensive education efforts are necessary to diminish poaching, especially pointless killings for sport.

### 3.5. Nepal

The mugger is one of the two species of crocodilians inhabiting Nepal, the other being the gharial.

#### 3.5.1. Historical Range and Status

Muggers were relatively common in the Terai region in the past [98]. They were also present in the West and East Rapti, Nariyani and Koshi river systems [99]. Known historical locations currently not inhabited by muggers are Gaidahawa Tal and Jagadishpur Reservoir [100]. By the early 1970s, the mugger population was limited to isolated remnants, due to overexploitation, intensive fishing and habitat degradation [99]. Surveys carried out during 1986–1988 estimated the minimum number of wild individuals at 100. The survey showed muggers had disappeared from eastern Nepal, their main stronghold being Royal Chitwan National Park. By 1994, Nepal’s population was estimated at 120–150 individuals [99].

#### 3.5.2. Current Range and Status

Currently, viable mugger populations are believed to inhabit Shuklaphanta, Bardiya and Chitwan National Parks, Koshi Tappu Wildlife Reserve and Ghodaghodi Lake complex (Figure 5) [5,100]. Baral and Shah, based on surveys, estimated Nepal’s total mugger population to be between 400 and 500 individuals, while the IUCN Red List assessment from the same year estimates 150–200 individuals [100]. Khadka et al. surveyed Chitwan National Park and its buffer zone for muggers, finding 245 muggers in 37 out of 58 wetland sites and in two out of three river systems [101]. They seemingly disappeared from the Reu river system within the last 20 years [99,101]. Annual surveys carried out from 2011 to 2017 in the Narayani and Rapti rivers pointed to an increase in the population of about 10% [102]. Seventy-three sites in lowland Nepal were surveyed in 2016, counting 704 muggers [103]. In Shuklaphanta National Park, mugger presence was recently recorded in the Chaudhar and Bahuni rivers, Rani Tal, Baba Tal, Solgaudi, the Mahakali River, Shikari Tal, Sundariphanta Khalla and Gobriaya Nullah [104]. A recent survey on the Ghodaghodi Lake complex counted 26 muggers [98].

#### 3.5.3. Habitat Description and Suitability

The Federal Democratic Republic of Nepal is a landlocked South Asian country situated on the Himalayas and the Indo-Gangetic Plain. It has a diverse geography, with extensive fertile plains as well as eight of the world’s highest mountains. Nepal is divided into three main topographic regions. The Terai is a flat, lowland river plain running the length of the southern border, is well suited for agriculture, and has a high human population density. The Siwalik (central hills) features deep valleys and steep slopes, mountain springs, terraced agriculture, and livestock farming. The High Himalayas run along the northern border and feature rangelands, glaciers and low human population density. Nepal experiences a wide range of climates varying from the sub-tropical to the Alpine type as the elevation varies from 64 to 8850 masl [105]. The country also experiences heavy rainfall during June to September due to the southwesterly monsoon, which accounts for 80% of the total rainfall [105]. There are four main river basins, originating in the Himalayas: Koshi, Narayani, Karnali and Mahakali, all of them susceptible to flooding. While, annually, Nepal receives an excess of rainfall, droughts occur seasonally [106]. All of Nepal’s muggers inhabit sites in Terai, the lowland region at the foot of the Himalayas, characterized by grasslands, savannah and swamps. Habitat suitability assessment in Napal is presented in Table 4.

#### 3.5.4. Threats

Habitat loss is believed to be the primary cause of reducing the number of muggers in Nepal. It has accelerated since the 1950s, when an intensive malaria eradication plan opened up the Terai region, previously densely inhabited by crocodiles, to human settlers [99]. Currently, approximately 40% of Nepal’s population lives in Terai [99]. Wetlands inhabited by muggers in Nepal are shrinking due to siltation and sedimentation, eutrophication, the deterioration of water quality and the construction of dams and barrages [99]. Severe eutrophication issues are caused by agricultural run-off and the invasion of alien plant species [100].

Intensive fishing causes a reduction in food resources as well as accidental drowning in fishnets. The use of gillnets and subsequent crocodile killings is still a major concern [100]. Crocodilian eggs are also believed by some locals to be an aphrodisiac, tempting them to poach eggs. While crocodiles used to have a place in Nepalese culture and religion, the growth of the human population caused them to be seen as a nuisance rather than holy animals. Ethnic groups rely heavily on fishing and aquatic resources (Sunaga, Khanwas, Mallahs, Bote, Mushahars, Bantar, Gongi, Mukhia, Dushad, Shani, Kewat, Danuwars, Darai, Kumal, Barhamus, Dhangar, Pode, Kushars and Majhi) [119].

Dam and barrage construction block migratory routes. Both hatchlings and adults are flushed below barrages during monsoon and have difficulties in returning to their initial locations [99]. This leads to isolation and possible loss of genetic diversity due to stochastic events.

Climate change presents a further risk, with flooding events doubling in frequency over the past decades [120]. According to the Climate Risk Index, Nepal ranks as the 10th most affected country in the world by climate change, particularly in terms of flooding events [120]. Incidences of dry spells, droughts, forest fires, heatwaves, flash floods, and disease outbreaks are increasing along with slow-onset risk [120]. Floods are of particular importance, as they destroy crocodile nests.

Muggers freely cross the Nepalese–Indian border, making their management an international issue, which complicates the implementation of conservation strategies [100].

#### 3.5.5. Captive Populations

Current information on captive stocks is unknown; however, by 1994, there were 99 muggers in captivity in Nepal [99]. We have found reports of muggers breeding in the Gharial Breeding Center in Kasara established in 1978, although we were unable to obtain the stock count.

#### 3.5.6. Reintroduction Efforts

A captive rearing program was established in 1978 at Kasara in Royal Chitwan National Park and at Bardiya National Park in 1982 [99]. Around 300 mugger eggs were collected, juveniles successfully reared and when they were around 1.5 m in size, they were released into Protected Areas between 1981 and 1994 [99]. Restocked muggers were not monitored.

#### 3.5.7. Other Conservation Efforts

Muggers became legally protected in Nepal in 1973 through the National Parks and Wildlife Conservation Act. Since its implementation, hunting for crocodiles is no longer permitted.

The majority of mugger habitats in Nepal are in Protected Areas, apart from the Lake Ghoaghodi Complex, although it is a recognized Ramsar site.

There is a considerable interest in wetland conservation. Wetland conservation policies have been present since the 1970s, finally being regulated by the release of the National Wetland Policy in 2012 [121]. Shortly after, The National Biodiversity Strategy and Action Plan (NBSAP 2014–2020) was released. Although the policies are pragmatic, their implementation faces considerable challenges, due to the poverty of communities reliant on wetlands.

#### 3.5.8. Prognosis

The mugger’s range in Nepal is limited; therefore, every site they inhabit should be monitored. It may be necessary to continue collecting eggs and rearing crocodiles due to the risk of flooding and water quality in some habitats (Shuklaphanta National Park, Ghodaghodi Lake) being insufficient to ensure the survival of hatchlings.

A disdain for crocodiles in local communities is a massive issue in mugger conservation. The dialogue between authorities and communities dependent on wetlands is severely limited. This issue will likely only become more severe; therefore, intensive education efforts and open communication are necessary.

### 3.6. India

India is inhabited by three species of crocodilians—the mugger, saltwater crocodile and the gharial.

#### 3.6.1. Historical Range and Status

Before the early 1970s, the mugger was considered to be common and widespread in India [122,123]. Its considerable decline in numbers caused by hunting for skins led muggers to disappear from numerous historical sites [38,122]. Many dam construction projects carried out between 1900 and 1950, as well as poaching for eggs and meat and accidental drownings in fishnets also considerably contributed to that decline [124].

In one of the Kaweri’s tributaries, the Kabini River (Karnataka), 23 muggers were counted during a survey in 1995 [125].

In Amaravathi Reservoir (Tamil Nadu), a survey from 1988 on nesting ecology recorded fewer nests (5) than the survey carried out in 1976 (11) [18].

By the 1980s, crocodiles were present, though scarce, in Neyyar Wildlife Sanctuary (Kerala). Thirty-six muggers were reintroduced into Neyyar Reservoir in 1983, which was followed by 36 reports of attacks on humans from 1985 onwards [126].

After the implementation of the Crocodile Conservation Project in 1975 in several states, through reintroduction programs, the establishment of Protected Areas and other efforts, the mugger population started growing once more.

#### 3.6.2. Current Range and Status

As of 2019, the total population size (non-hatchlings) is estimated at 4000 individuals with viable populations inhabiting at least 12 states (Figure 6) [7].

Muggers inhabit the Kaweri River (Karnataka, Tamil Nadu). Surveys from 2019 carried out in Melagiris (a southern stretch of the Kaweri) counted 54 crocodiles on a 24 km stretch of the river, which equates 2.25 muggers per kilometer [127]. In that survey, both eggs and hatchling were observed.

Surveys from 2014 to 2016 on the Moyar River (Karnataka, Tamil Nadu) revealed 98 known spots of crocodile activity on a 102 km stretch of the Moyar and its tributaries and density ranging from 0.41 to 0.51 crocodile per km, depending on the season [128]. A concurrent survey on a 26 km stretch of the Moyar counted 81 muggers, meaning a density level of 2.89 muggers per km [129]. Such a large discrepancy most likely resulted from the fact that the former survey covered tributaries and stretches of the river at high elevation, while the latter focused on a lower elevation stretch of the Moyar, where crocodiles are more abundant.

Amaravathi Reservoir (Tamil Nadu) is believed to house the largest population of muggers in India [18]. Vasudevan reported anecdotal evidence of the population falling in number, although no research took place to prove the decline [18].

In Neyyar Reservoir (Kerala), surveys from 2001 estimated the mugger population at 10 to 16 individuals [130].

According to the Parambikulam Tiger Reserve Management Plan for the period of 2011–2020, muggers are frequently sighted in the reserve, especially on the banks of the Parambikulam and Thunacadavu Reservoirs (Kerala) [131].

In Gujarat, there is a considerable mugger population, spread across the Kheda, Anand and Vadodara districts. A survey carried out in 2011–2012 showed at least 470 muggers in 32 water bodies across these three districts and breeding was reported from 11 sites [132]. Vyas reported the highest density in the Vishwamitri River [132]. The following survey on Vishwamitri within Vadodara in 2015 reported a density of 9.1 individuals per km in the city’s boundaries [132]. Between 2013 and 2015, muggers were recorded at 27 locations within Kheda and Anand, with an overall density of 14.31 individuals per 100 km^2^ [133]. Of these observations, 29% were recorded within the village of Deva [133]. In Pond Deva, a study from 2018 on basking behavior reported 40 muggers in the pond [134].

Muggers are also present in Chambal’s (Madhya Pradesh, Rajasthan, Uttar Pradesh) main channel and its tributaries, occasionally spotted in the Banas River (Rajasthan) and more common in the Parvati River (Madhya Pradesh), with surveys reporting 83 individuals in 2015 and 66 in 2016 [135].

In Odisha, muggers inhabit Similipal Tiger Reserve, where it was extirpated around the 1980s and later returned thanks to successful reintroduction [136]. Surveys in the Reserve carried out in 2019 counted 82 muggers, their majority found in the West Deo river system [136].

In Uttar Pradesh, Dudhwa Tiger Reserve is known to house muggers. A survey from 2013 that lasted for over two days counted 20 crocodiles in the Girwa River flowing through the reserve [137].

In Uttarakhand, muggers are present in waterbodies across the Terai Arc Landscape. According to studies carried out in 2014–2017, crocodiles were present in 10 sites in the Western Cricle and the total estimated population size was 281 individuals [138]. Out of those 10 sites, most individuals were recorded in Kakra Canal, Nanak Sagar Dam, Dhora Dam and Sharda Dam.

While we found no surveys of the following locations, the presence of crocodiles was recently confirmed through news reports of human–crocodile conflict in rivers: Bhadra (Karnataka), Kali (Karnataka), Kollidam (Tamil Nadu), Bhavani (Kerala, Tamil Nadu), Narmada (Gujarat), Krishna (Maharashtra, Karnataka, Andhra Pradesh, Telangana), Godawari (Maharashtra, Telangana, Andhra Pradesh) as well as from Kaliyasot Dam (Madhya Pradesh).

#### 3.6.3. Habitat Description and Suitability

India is a large South Asian country lying on the Indian subcontinent. Due to India’s size, the climate varies greatly between regions. For the most part being a tropical country, it reaches the temperate belt north of the tropic of Cancer. It has two distinct seasons, the dry season and monsoon season, apart from desert regions that do not experience monsoon rains. Temperature ranges from −40 °C in Jammu and Kashmir to 55 °C in Rajasthan’s deserts. Similarly, the average rainfall varies greatly, from 20 cm a year in desert regions to over 1000 cm in most areas. India is divided into 10 biogeographic zones. Out of those 10, the Trans Himalaya and Himalaya region, collectively covering 12% of the country and Northeastern region (5.2%) are not suitable habitats for crocodiles. The largest ecoregion is the Deccan Peninsula (42% of India). The elevation on the Deccan Peninsula is between 300 to 750 masl. It is located in the rain shadow of the Western Ghats and its climate is semi-arid or semi-evergreen, depending on the location. While the zone is varied, it is mostly covered by deciduous forests, with the addition of thorn forests and scrublands. The semi-arid zone and desert zone cover 16.6% and 6.4% of India, respectively. The elevation in these zones varies from 20 to 450 masl. Apart from deserts, these regions are covered by grasslands and shrubs and even by deciduous forests and tropical thorn forests in the semi-arid zone. The Gangetic Plain covers 10.8% of India. These large flood plains are mostly agricultural and characterized by a dense human population. Western Ghats cover 4% of India. It is one of India’s largest tropical evergreen forests with a high number of endemic species and high biodiversity (one of 25 world biodiversity hotspots). The region consists mostly of hills, where the majority of rivers of South India have their beginning. The coasts (2.5%) and islands (0.3%) consist mostly of sandy beaches and mangrove forests and are inhabited mainly by saltwater crocodiles, more adapted to brackish and saline habitats.

Habitat suitability assessment in India is presented in Table 5.

#### 3.6.4. Threats

Due to the vast range of muggers in India, as well as climate and cultural variability across the country, each site faces different challenges. The two main threats can be categorized into habitat degradation and human–crocodile conflict.

Habitat degradation is an issue faced all over the world, but a few countries house nearly 20% of the world’s population. As of 2022, India had a population of a staggering 1.42 billion people and the United Nations predicts that the growth will not reach its peak until 2064 [182]. Resources are already strained, especially water and agricultural land availability, followed by wood and minerals.

India is in a water crisis. The Union Water Resources Ministry projects that the total water demand will increase by 78% by 2050 [183]. India’s demand for water is expected to exceed the country’s replenishable water capacity by 2025 [183]. As India is the major exporter of goods across the world, it also indirectly redistributes water resources by exporting products that require considerable amounts of water to produce. With glaciers in the Himalayas retreating and rising sea levels caused by climate change, water stress will continue to rise in severity [184].

The growing population not only increases the demand for natural resources, it also directly correlates to pollution. Untreated sewage, industrial waste and agricultural flows find their way into most of India’s river basins. A proportion of 70% of the Indian states treat less than half of their waste discharge, resulting in the introduction of potentially harmful pathogens into natural habitats [183]. Water depletion, either due to a lack of water or its contamination, affects two-thirds of India’s districts [185]. Contamination is so severe that 70–80% of the country’s overall disease burden is caused by polluted water [186].

India’s rivers are also heavily altered, either by interference into the riverbed and riverbanks or by the construction of dams. Dams obstruct the migration of wildlife, disturb the natural river flow and may directly or indirectly change the physical and chemical characteristics of water, such as turbidity, temperature and oxygen demand.

Despite water shortages, due to India’s geographical structure, parts of the country are highly susceptible to flooding, further deteriorated by the mismanagement of sewers in urban areas and siltation and erosion caused by human activities [187]. This has caused a rise in flooding events in India in the last century and the occurrence of floods in areas previously considered to not be flood-prone [187]. Flooding, alongside drought, is considered to be the most prevalent natural threat to crocodilians.

Apart from water-related issues, natural habitats are degraded by conversion into agricultural and industrial land. While it would be unrealistic to aim for the cessation of land conversion in the face of India’s population growth, Protected Areas at least should be exempt from land conversion. Unfortunately, encroachment is an issue in many Protected Areas. Villages of unambiguous legal status exist in two-thirds of Protected Areas in India and 20% of Protected Areas experience physical confrontation between management and locals [171,188]. Satellite imaging has detected that 31,677 ha of Protected Areas in Karnataka alone are occupied by agriculture, horticulture, construction and pasture [188]. The rights of indigenous people and tribes present a separate subject, as their customary rights for land and resources are protected by international human rights laws and scheduled tribes and other traditional forest dwellers are protected under the 2006 Forest Rights Act. Indigenous communities are often vital for conservation, as their traditional knowledge of the land and resources and willingness to oppose external threats prove invaluable [189].

Habitat degradation and dam construction contribute to habitat fragmentation. Difficulty in migration and dispersion hinders gene flow among subpopulations. Hindered gene flow may result in the loss of alleles due to genetic drift and eventually in inbreeding depression [190,191]. This in turn increases the risk of species extinction due to reduced adaptability to environmental changes and stochastic effects [192,193]. It also forces crocodiles to migrate through roads and railroads, resulting in vehicular collisions. Vyas et al. report 15 cases of crocodile–vehicle collisions in the years 2021–2022 in Gujarat alone [194].

Fear of injury and loss of life and depredation of livestock is affecting people’s support for conservation efforts and their willingness to limit activities that are harmful to wildlife. As freshwater habitats are indispensable both to humans and crocodiles, they compete for water, space on banks and fish stock. In India, there are cases where crocodiles inhabit big cities, as in Vadodara (Gujarat). Conflict is apparent in the areas of high density of both human and mugger populations, in communities reliant on fishing and in areas where crocodiles were restocked in sub-optimal habitats, such as the Neyyar dam in Kerala or the Godawari delta in Andhra Pradesh.

In Gujarat, between 1960 and 2013, a total of 64 attacks were recorded, although the actual number may be higher because not all of them are reported [195]. Most of these attacks occurred in central Gujarat in the basins of Vishwamitri and Narmada [195]. Vyas and Stevenson (2017) also noted a gradual increase in attacks in the region [195]. In the Kaweri delta in Tamil Nadu, 19 attacks were reported between 2009 and 2019 [141], and in Maharashtra, 16 fatal attacks between 2003 and 2017 [168]. As the human population rises, habitats degrade and fish stock depletes, these attacks would likely become more prevalent. The lack of education, illiteracy and poverty may likely further contribute to the deepening of human–crocodile conflict. People sharing spaces with crocodiles often are not aware of the foraging times, breeding seasons and other factors increasing the risk of attacks. Still, it should be noted that crocodile attacks on humans are far less frequent than, for example, fatalities from snake bites in India. 

The depredation of livestock is another issue fueling the human–crocodile conflict and, as is the case with attacks on humans, it is likely to rise in severity in the future. Approximately 15,730 cases of livestock attacked by wildlife were reported from 18 states, costing them over one million USD in compensation [118].

Human–crocodile conflict goes both ways, which is visible in the numbers of deaths in fishing nets. De Silva and Lenin [1] report accidental drowning in fishing nets as one of the major threats to muggers. India is the third largest fish producer in the world, contributing approximately 7% of the global fish production. Fish production is another developing sector in India, continuously growing over the last few decades [196]. Around 74% of fish products come from inland water basins [196]. As of this writing, gillnets have not been banned in India.

#### 3.6.5. Captive Populations

In response to the considerable loss of the crocodile population up until 1970, the United Nations Development Programme/Food and Agriculture Organisation with the cooperation of the Government of India launched the Crocodile Breeding and Management Project in 1975. 

The objectives of the project included the collection and incubation of eggs and establishing a network of rehabilitation centers. After its implementation, 16 rehabilitation centers were established across India and by 1984, 10 of these centers managed to successfully breed muggers [121]. By 1990, the CSG reported an excess of 12,000 crocodiles in captivity [197]. In 1990, Madras Crocodile Bank alone kept 2842 individuals [198]. By 1994, the Ministry of Environment appealed to captive centers to stop breeding muggers [39]. As of 2021, a total of 1968 are held in nine captive breeding centers and zoos [66].

Unfortunately, despite large numbers of captive muggers, the government vehemently opposes farming. Farming could potentially allow captive crocodiles to be sustained without financial loss and further integrate local communities into conservation, by presenting crocodiles as a viable livelihood source.

#### 3.6.6. Reintroduction Efforts

Based on reintroduction program reviews, research papers and reports from the Crocodile Specialist Group Newsletter, we found records of 1607 muggers released into the wild in the locations listed in Table 6 [122,126,199,200,201,202,203,204]. Breeding centers were listed whenever the information was available.

In locations where reintroduction success was measured, the level of survivability averaged at around 80% [199]. In areas such as Neyyar Wildlife Sanctuary, the deepening of human–crocodile conflict was observed as a result of reintroduction [126].

FAO collaboration ended in 1982 and reintroductions were limited due to the rising population of muggers in India and the shortage of suitable habitats.

#### 3.6.7. Other Conservation Efforts

The principal legislation meant to protect wildlife in India is the Wild Life Protection Act from 1972, where muggers are listed in Schedule I, providing them with absolute protection in the eyes of the law. In response to the considerable loss of the crocodile population until 1970, the United Nations Development Programme/Food and Agriculture Organisation, with the cooperation of the Government of India, launched the Crocodile Breeding and Management Project in 1975. 

In accordance with the project, 11 crocodile sanctuaries were established [198]. As of 2024, 1014 Protected Areas were established in India overall, covering 17,516,942 ha (5.32% of India).

To alleviate human–crocodile conflict, compensation is paid for loss of life, injury and depredation of livestock. Their level is based on state law and the effectiveness of governmental bodies to compensate varies from state to state. Not all states list muggers as a species under compensation policy. Unfortunately, compensation does not reach the actual market value of livestock and the process of obtaining compensation is lengthy and difficult, especially considering the illiteracy rate in poor regions [205]. It is also difficult to assess whether compensation fees for human injury and death have a positive impact on human attitude toward crocodiles.

Nuisance crocodiles are often translocated. Within Vadodara (Gujarat), 365 individuals were translocated between the years 2007 and 2017 [195]. Crocodiles are often released into convenient nearby sites, without considering their territorial nature and migration capabilities [195]. Large nuisance crocodiles are sometimes translocated into captivity and never return to the wild, therefore eliminating the largest individuals from the gene pool [195]. There is no monitoring of translocated individuals; therefore, it is impossible to know whether crocodiles return to their original territory or create issues in the place of translocation. There is a need for adopting a state- or nation-wide protocol for crocodile translocation.

#### 3.6.8. Prognosis

The possibility of muggers’ survival in India vary greatly from site to site. Although it is unlikely muggers will disappear from India in the near future, almost all threats seem to only gain in severity across the years and require monitoring to ensure a swift response.

The better management of surface water is necessary for the preservation of crocodiles, as well as other wildlife and humans. Controlling sewage and industrial waste discharge should be an absolute priority.

Education is necessary to mitigate human–crocodile conflict. Traditional methods of preventing attacks, mainly through caution, the avoidance of water bodies during foraging times and their partial covering with vegetation banks during the breeding season should be employed by the general public. The translocation of troublesome crocodiles is not enough to prevent the inflammation of human–crocodile conflict and adherence to protocols is required.

## 4. Results

Based on the literature review, we have summarized the status, threats and chances of survival in the Table 7.

Changes from the historic ranges were noted where possible, although there is limited concrete information of the historic range in most countries. The population size trend was determined by surveys and population estimates from the years 2000 to 2023. It is important to note that due to the complexity and severity of threats to muggers, the rising population may not directly correlate to the chance of survival.

The habitat suitability was summarized by assigning each described habitat a score from 1 to 3 (1—good, 2—moderate, 3—bad), based on the severity of threats and the current status of habitats, as they pertain to the chances of mugger survival. The habitats in each country were compiled and the values presented in the table below correspond to the mean values for each country.

The major threats are listed in order of severity for each country, based on expert reports and the number of habitats affected. The chance of survival was determined based on expert reports, the severity of threats and the difficulty and complexity of mitigation efforts (for example, a program for collecting eggs and rearing hatchlings to mitigate flooding is easier to establish than mitigating droughts in countries experiencing water shortage).

## 5. Conclusions

Based on our findings, we can conclude that the chance of preserving muggers varies greatly among countries and habitats across its range. We believe the main stronghold of muggers to be India, although water shortage in Iran will likely have a great impact on crocodiles in the not-so-distant future. While each site faces threats of different characteristics and severity, we can distinguish two main types of threats—those associated with habitat degradation and with human–crocodile conflict. Habitat degradation is an issue in every country, stemming from several factors: encroachment, conversion to agricultural and industrial land, climate change, water shortage, water pollution and integration into natural river banks. Human–crocodile conflict is generally more site-specific and is a major issue in large cities, rural areas and in Protected Areas inhabited by indigenous people and stems mostly from attacks on humans and livestock for people and general disturbance and the use of fishnets for crocodiles. These threats are complex and require a multidisciplinary approach and comprehensive management strategies.

## Figures and Tables

**Figure 1 animals-14-00691-f001:**
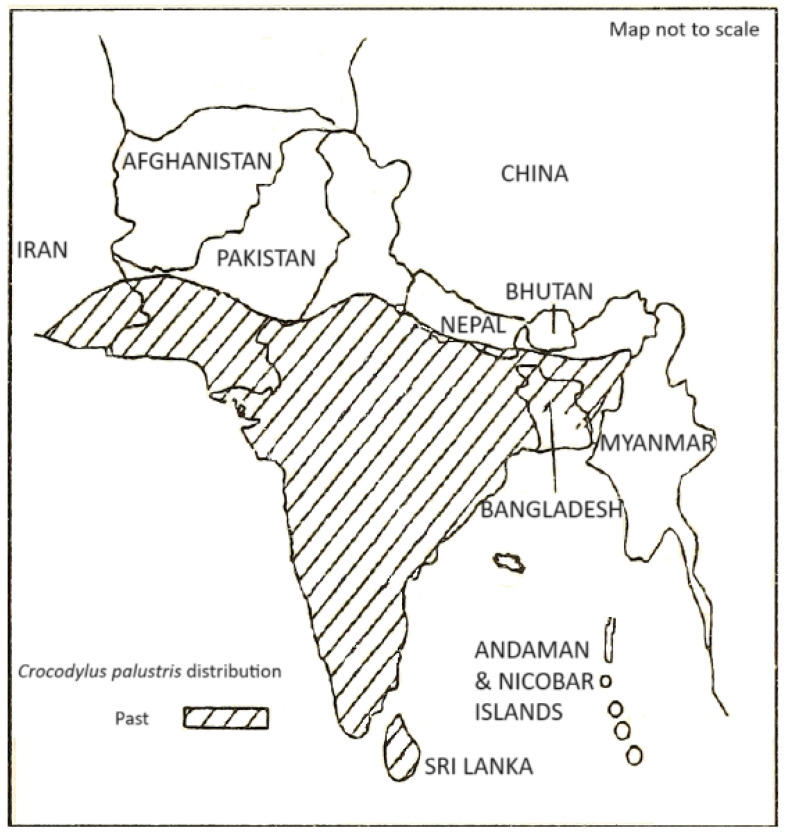
Map of mugger historical distribution [5].

**Figure 2 animals-14-00691-f002:**
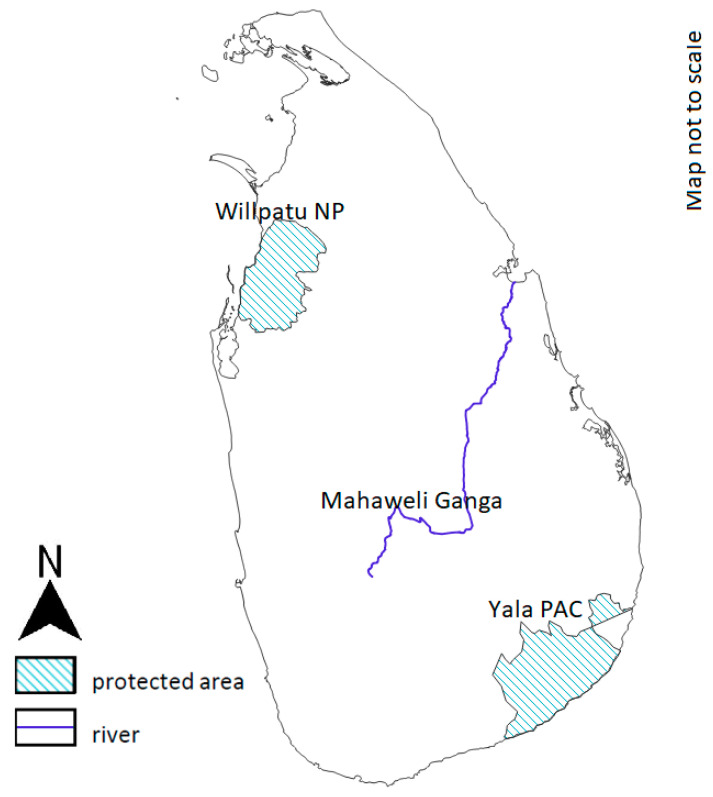
Map of the assessed habitats of the mugger in Sri Lanka.

**Figure 3 animals-14-00691-f003:**
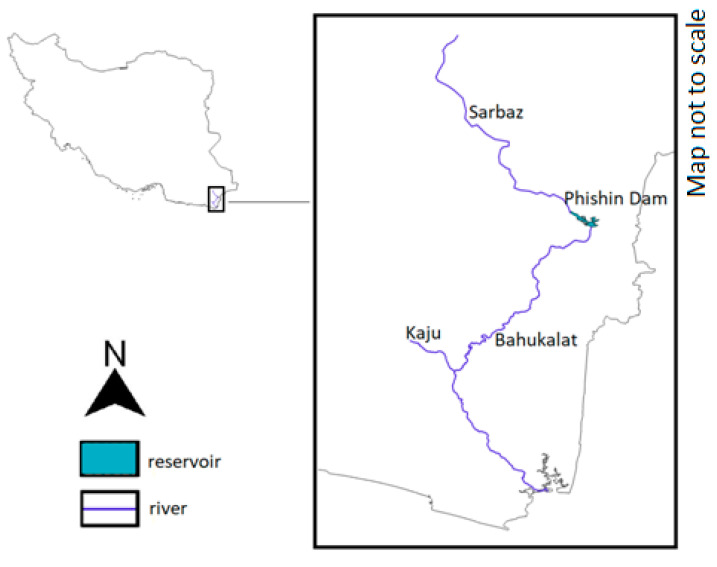
Map of the assessed habitats of the mugger in Iran.

**Figure 4 animals-14-00691-f004:**
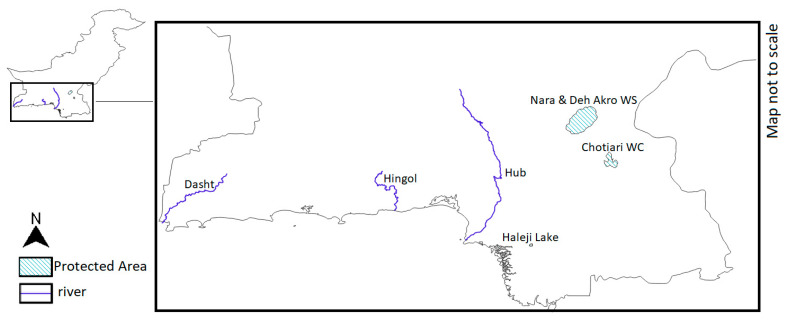
Map of the assessed habitats of mugger crocodile in Pakistan.

**Figure 5 animals-14-00691-f005:**
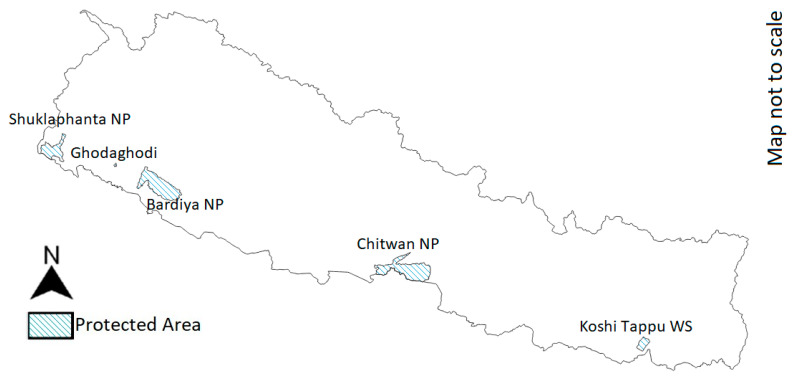
Map of the assessed habitats of the mugger in Nepal.

**Figure 6 animals-14-00691-f006:**
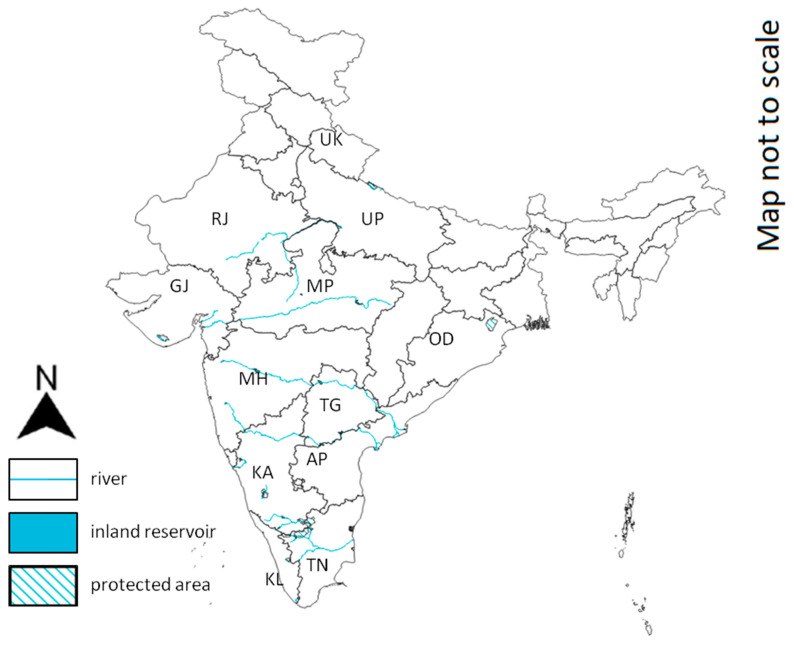
Map of the assessed habitats of the mugger in India.

**Table 1 animals-14-00691-t001:** Habitat suitability in Sri Lanka.

Habitat	Legal Status	Elevation	Surface Water Availability	Water Quality	Nesting and Basking Site Availability	Interactions with Humans	Notes	Suitability
Mahaweli Ganga	Flows partly through the Flood Plains National Park [33]	>60 masl	Perennial river	Contaminated with household and industrial waste, siltation	Riverbanks partially altered by sand mining and transport [34]	N/A	Large dams along the river	Moderate
Willpatu National Park	Protected Area	>240 masl	Numerous flood plains, swamps, mangroves, estuaries, villus, perennial and seasonal rivers [35]	Good to moderate	N/A	Encroachment, illegal timber extraction [36]		Good
Yala Protected Area Complex	Protected Area	>125 masl	Streams, tanks, water holes, rock pools and lagoons, some drying during the dry season	Good to moderate	N/A	Encroachment, poaching and free-range livestock [37]		Good

N/A—data necessary for assessment not available.

**Table 2 animals-14-00691-t002:** Habitat suitability in Iran.

Habitat	Legal Status	Elevation	Surface Water Availability	Water Quality	Nesting and Basking Site Availability	Interactions with Humans	Notes	Suitability
Kaju River	Gando Protected Area	N/A	Susceptible to droughts and overflows [63]	N/A	Inadequate nesting sites along big dams [58]	Loss of livestock to crocodiles	Ziridian Dam on the river serves as a refuge during prolonged droughts [58]	Moderate
Sarbaz River	Gando Protected Area	N/A	Susceptible to drought and overflows [63]	High concentration of lead in sediments [64]	Inadequate nesting sites along big dams [58]	Loss of livestock to crocodiles	Phishin Dam on the river holding one-third of the estimated Iranian mugger population [58]	Moderate
Bahukalat River	Gando Protected Area	N/A	Perennial river	High concentration of iron, mercury and lead in blood serum of resident crocodiles [65]	N/A	Loss of livestock to crocodiles	Pollution caused by discharge of industrial and municipal effluents and mine drainage	Moderate

N/A—data necessary for assessment not available.

**Table 3 animals-14-00691-t003:** Habitat suitability in Pakistan.

Habitat	Legal Status	Elevation	Surface Water Availability	Water Quality	Nesting and Basking Site Availability	Interactions with Humans	Notes	Suitability
Deh Akro II Wildlife Sanctuary	Protected Area	60–80 masl	36 lakes varying in size (40 to 750 ha) and depth (2 to 15 m)	High levels of chloride, sulphate, calcium, bicarbonate and carbon trioxide, high salinity and TDS factor, eutrophication [84]	Suitable nesting materials	Drownings in fishnets, noise and light disturbance [79]	Siltation is mitigated by annual desiltation of the Nara and Jamrau canals carried out by the government. Despite this, due to water shortage, climate change and eutrophication surface water availability and quality steadily decreases.	Moderate/bad
Nara Desert Wildlife Sanctuary	Protected Area under risk of overexploitation	60–80 masl	More than 200 small lakes, irrigation canals in the northern part [85]	Moderate, varies greatly across lakes, should be monitored [85]	Suitable nesting materials	Gas exploration, road construction, wood cutting, waste dumping [85]	As of 2012, crocodiles were present in at least 19 lakes [80]. Surface water availability has steadily decreased.	Moderate
Chotiari Wetland Complex	Protected Area	60 masl	Wetland comprises many freshwater and brackish lakes [86]	Elevated levels of chloride, calcium, magnesium, sodium, potassium, sulphate, bicarbonate, high salinity, eutrophication [80]	Limited nesting sites [80]	Drownings in fishnets, human disturbance on potential nesting sites [80]	As of 2015, presence of crocodiles was confirmed in 16 lakes [80]. Water pollutants seem to be introduced mostly by agricultural runoff [80]. Surface water availability and quality have steadily decreased.	Bad
Dasht River	Not under legal protection	Approximately 78 masl	Seasonal river	Elevated levels of fluoride [87]	N/A	Livestock losses to crocodiles, preventative killings [4]		Moderate
Hub River	Hub Dam Wildlife Sanctuary	90–141 masl	Perennial river (under normal conditions)	Contaminated with fecal matter and heavy metals [88]	N/A	N/A		Moderate
Hingol River	Partially flowing through the Hingol National Park	N/A	Ephemeral river [89]	Contaminated with fecal matter and lead [89]	N/A	N/A		Data deficient
Haleji Lake Wildlife Sanctuary	Protected Area	N/A	Lake covering 6.58 km^2^	Good to moderate depending on the season [81]	N/A	Drownings in fishnets, seven injured and three killed by crocodiles, pointing to significant human–crocodile conflict [80]		Moderate
Mangho Pir	Shrine	Approximately 8 masl	Lake 5–6 m deep, fed from a nearby stream [80,90]	Water quality moderate, actively maintained [80,90]	N/A	Human disturbance (tourist spot)	Lake protected by a brick wall. Crocodile density became an issue, as crocodiles are often observed fighting over space and food [91].	Moderate

N/A—data necessary for assessment not available.

**Table 4 animals-14-00691-t004:** Habitat suitability in Nepal.

Habitat	Legal Status	Elevation	Surface Water Availability	Water Quality	Nesting and Basking Site Availability	Interactions with Humans	Notes	Suitability
Chitwan National Park	Protected Area under risk of overexploitation	110–850 masl	Three main rivers draining the park, 58 lakes and wetlands, rivers experience severe flooding [101,107,108]	Siltation	Flooding of nesting sites	Poaching, intensive fishing, preventative killings, (approximately five muggers annually are killed) [101,107]	Invasive plant species and siltation cause wetlands to steadily shrink. Maskey proposed that flooding is the main factor limiting crocodile population in the Naryani river [108]. Rivers are inhabited by the gharial and river dolphin [107].	Moderate
Shuklaphatna National Park	Protected Area under risk of overexploitation	174–1386 masl	Three major rivers flow through the park	The water quality parameters (dissolved oxygen, total hardness, free carbon dioxide, orthophosphate, biological oxygen demand and ammonia) of Rani Tal wetlands in Shuklaphanta exceeded the normal range to support the mugger [109]	N/A	Poaching, preventive killings, feral animals, habitat degradation and pesticide use in the buffer zone [104]		Moderate
Bardiya National Park	Protected Area	152–1564 masl	Two major rivers drain the area; due to water scarcity in dry season, 50 man-made ponds have been provided to wildlife, though in monsoon season the region experiences flash floods [110]	Pollutants were detected in Babai river flowing through the Bardiya National Park [111]	Overexploitation of river bank materials [110]	Human–wildlife conflict is a major issue, but local communities are more focused on crop loss to elephants and rhinos and tiger attacks than crocodiles [112]	The park is also inhabited by the gharial.	Moderate
Koshi Tappu Wildlife Reserve	Protected Area under risk of overexploitation	85–90 masl	Perennial river Koshi Tappu, overflowing in the monsoon season	N/A	N/A	Thousands of locals enter Koshi Tappu WR daily to collect firewood, grasses, timber and fish, causing a significant human–wildlife conflict [113,114]. Human–crocodile conflict in the area stems mostly from depredation of fish in private ponds [114]	Study on land cover changes between 1976 and 2010 show a decline in river and swamp cover of about 17% [115]. There is a considerable conflict of interest between local communities and Wildlife Reserve personnel and Royal Nepalese Army, tasked with preventing encroachment [116]. Planned construction of a high dam on the Koshi river is a major concern, since the dam would drastically change the relatively undisturbed river basin [117]. As of this writing, the project remains in planning stage. The reserve is also inhabited by the gharial and the Ganges river dolphin [113]	Moderate
Lake Ghodaghodi	Lake Ghodaghodi Complex Ramsar site	205 masl	Lake Ghosaghodi covers an area of 150 ha; 14 other lakes in vicinity form the Ramsar site. The area faces water shortage in dry seasons and severe floods during monsoon season [118]	Ghodaghodi lake was characterized as hypertrophic (due to high phosphate levels), polluted by high nutrient deposition from decaying aquatic flora [118]	N/A	Wetlands are threatened by poaching, sedimentation, settlement development, invasive species and drainage/reclamation of land for agricultural purposes [119]. It is also a holy site for indigenous Tharu community, celebrating festivals (Agan Panchami) by entering the lake [119]	Threat assessment carried out by Lamichhane et al. listed illegal fishing and habitat modification as most prevalent threats to mugger [98]. They reported human–crocodile conflict to be on a manageable level.	Moderate/bad

N/A—data necessary for assessment not available.

**Table 5 animals-14-00691-t005:** Habitat suitability in India.

Habitat	Legal Status	Elevation	Surface Water Availability	Water Quality	Nesting and Basking Site Availability	Interactions with Humans	Notes	Suitability
Kaveri River (Tamil Nadu, Karnataka, Kerala).	Part of the Kaweri flows in the boundaries of Cauvery Wildlife Sanctuary and Ranganathittu Bird Sanctuary.	41.88% of the basin lies below 400 masl.	Perennial river, multiple water bodies are located in the basin: over 42,000 lakes, ponds and reservoirs.	200 km stretch of river non-complying to the Water Quality Criteria [138].	The basin faces a serious issue of river bank erosion, with high slopes along most of its run.	There were 20 reported attacks on humans by muggers from 2009 to 2019 in Kaweri’s delta, seven of those being fatal [139].	There are 96 dams of varied size along the Kaweri. Parts of the river are located in regions with high human density; the human population growth was estimated at 17.25% in the region. There are a number of industries in the basin, including the textile industry, cement factories and metal plants.	Moderate.
Kabini River (Kerala, Karnataka).	Not under legal protection.	N/A	Perennial river.	5 km polluted stretch of the river [140]. Heavy metal pollution in sediment samples, namely manganese, copper and zinc [141].	Siltation is observed around the river banks [142].	The river is mainly utilized as water source for crops and livestock, with intense fishing in Kittur village only [142].	There is one large dam—Kabini dam—creating a vast reservoir. Currently, the Kabini river is a candidate for developing an Inland Water Transport route, which would require human interference into river banks [142]. The main pollution sources are sewage discharge and municipal solid waste in Nanjanagud [143].	Moderate.
Kollidam Canal (Tamil Nadu).	Not under legal protection.	N/A	Perennial canal, high risk of flooding due to sediment deposition from sand mining operations	Heavy metal contamination (copper and cadmium) in estuarine sediments and five species of freshwater fish [144].	Severe river bank erosion due to both legal and illegal sand mining [145].	There were 20 reported attacks on humans by muggers from 2009 to 2019 in Kaweri’s delta, seven of those being fatal [139].		Bad.
Bhavani River (Kerala, Tamil Nadu).	Not under legal protection.	200–3000 masl.	Perennial river, threatened by decreasing groundwater levels [146].	Heavy metal contamination, fluoride concentration exceeding permissible limit on a 60 km stretch of Bhavani [146].	N/A	N/A	Contamination results from considerable industrialization of the region, with 400 units of paper, dyeing, sugar and bleaching industries, that both use Bhavani’s water and expel waste into the river [147,148]. Tamil Nadu Pollution Control Board recognizes discharge of untreated domestic sewage as the main pollution source [149]. There are two dams on Bhavani, out of which Bhavani Sagar is known to house crocodiles. It is one of the largest earthen dams in the world and creates the second largest reservoir in Tamil Nadu, with capacity of 928,000,000 m^3^ [148,150].	Moderate.
Amaravathi River (Tamil Nadu).	Not under legal protection.	40–500 masl in the plains [151].	Perennial river.	Heavily polluted; Ahamed and Loganathan detected lead, cadmium and nickel that exceeded permitted levels for drinking water, and categorized gathered water samples as semi-critical in water quality [152].	N/A	N/A	Pollution is the major issue, due to textile and bleaching industry units located along the river [151,152]. Amaravathi Dam is believed to house one of the largest populations of muggers in India [18].	Moderate.
Moyar River (Karnataka, Tamil Nadu).	Partially flows through Mudumalai Tiger reserve, Sathyamangalam Tiger Reserve and Nilgiri North and South Divisions.	250–2054.	Perennial river.	Stretches of river are under considerable eutrophication.	N/A	Agricultural runoff, hydroelectric projects, unrestricted fishing activities (including occasional use of dynamite), pesticides and spilling of motor oil [128].		Moderate.
Bhadra River (Karnataka)	Falls under Bhadra Wildlife Sanctuary and Bhadra Tiger Reserve.	N/A	Perennial river.	There is an identified 10 km stretch of polluted water starting at Holehunnur and ending at Bhadravathi [140,153].	N/A	N/A		Data deficient.
Kali River (Karnataka).	Not under legal protection.	N/A	Perennial river.	Identified as polluted on a 10 km stretch of Dandeli, due to sewage discharge [154].	N/A	The river is utilized for tourism and recreational purposes and fishing. There are five reports of mugger attacks on humans in Kali’s vicinity in the last decade. Rising frequency of attacks on humans is likely caused by construction on the river in Dandeli.		Moderate.
Neyyar Wildlife Sanctuary (Kerala).	Protected Area.	100–1868 masl.	Reservoir.	Moderate [155].	Erosion of reservoir banks [156].	36 muggers were reintroduced into Neyyar reservoir in 1983, which resulted in considerable fueling of human–crocodile conflict, due to 30 reported attacks from 1983 to 2001 [126,157]. Locals also reported frequent attacks on livestock [126]. Local attitude towards crocodiles was reported as hostile in 2001 [126].	Vijayasoorya et al. reported degradation of forest cover in the sanctuary, which, according to landscape analyses, declined by 10% between 2011 and 2015 [158].	Moderate.
Parambikulam Tiger Reserve (Kerala).	Protected Area.	300–1438 masl.	Apart from Parambikulam, there are two other man-made reservoirs and two rivers flowing through the reserve.	N/A	N/A	N/A		Data deficient.
Vishwamitri River.	Not under legal protection.	N/A	Seasonal river, susceptible to flooding.	The river is heavily polluted due to sewer and industrial waste disposal and solid waste dump sites [159].	Suitable nesting sites available [160].	Numerous attacks on livestock and domestic animals suggest dependency of crocodiles on livestock as food source [131]. They are also observed scavenging on dumping sites and on carcasses, possibly illegally dumped into the river by hospitals [131]. From the period of 2014–2015 alone, 24 attacks on humans were reported, 12 being fatal [161].	The river is surrounded by urban, rural and industrial landscape, the historic river being converted into a sewer [159].	Moderate.
Narmada River (Madhya Pradesh, Gujarat).	Not under legal protection.	N/A	Perennial river.	160 km stretch of polluted river in the boundaries of Madhya Pradesh [100].	N/A	Indian media report five attacks on humans in last decade.	There are 21 major dams on the river [162].	Moderate.
Kaliyasot Dam (Madhya Pradesh).	Not under legal protection.	N/A	Reservoir.	Moderate, high turbidity and alkalinity, high nitrate and sulphate levels [163,164].	Suitable nesting sites.	Human disturbance and frequent encroachment.	Kaliyasot dam is located within Bhopal, a city with a population of two million. According to Silawat and Chauhan, the reservoir is under high environmental stress due to human encroachment, siltation, high macrophytic growth and sewage discharge [163].	Moderate/bad.
Godawari River (Maharashtra, Telangana, Andhra Pradesh, Chhattisagarh, Odisha).	Not under legal protection.	N/A	Perennial river.	N/A	N/A	N/A	The river is under environmental stress due to rapid urbanization, building of dams, destruction of riparian vegetation, unregulated construction along river banks a sewer discharge [165].	Data deficient.
Krishna River (Karnataka, Maharashtra, Andhra Pradesh).	Not under legal protection.	300–600 masl on the plateau [166].	Perennial river.	Central Pollution Control Board deemed that more than half of the river (750 km) should be considered polluted [140].	River banks susceptible to land sliding [167].	Atigre reports 16 attacks on humans and 62 attacks on cattle from 2003 to 2017 in Sangli district alone [168].	As of 2014, total of 660 dams were built on the Krishna [166]. In 2014, there were 11 894 industries in the basin, including sugar factories and sand mining operations [169]. MITRA recognizes the major sources of pollution to be disposal of untreated sewage, industrial effluent, agricultural runoff, religious waste, disposal of municipal solid waste, biomedical waste, hazardous waste and sand mining [169].	Moderate/bad.
Silimpal Tiger Reserve (Odisha).		500–600 masl.	Numerous perennial streams forming three main river systems [170].	N/A	N/A	Native tribes live in the vicinity of the reserve, with 65 villages falling into its boundaries, highly dependent on resources provided by the forests [171,172]. Poaching has become a big problem, especially during Akhand Shikar, a ritual mass hunting event. Losses in livestock to predators are an issue in the area.		Moderate.
Chambal River (Madhya Pradesh, Rajasthan, Uttar Pradesh).	Partially flowing through National Chambal Sanctuary.	111–843 masl.	Perennial river.	Good [173,174,175].	Suitable nesting sites.	Overfishing, drownings of crocodilians in fishing nets and illegal sand mining. Limited human–crocodile conflict in the area, mostly due to depredation of livestock.		Good.
Banas River (Rajasthan).	Not under legal protection.	176–1291 masl.	Seasonal river.	There is a 60 km long patch of the river polluted with chloride, nitrate and fluoride [176].	N/A	N/A	The Banas River has dried out since the Bisalpur Dam was completed in 1999, with restricted flow outside of monsoon season [134]. A proportion of 90% of Rajasthan was experiencing water stress as of 2014. There are occasional reports of gharial sightings in Banas [134].	Moderate.
Dudhwa Tiger Reserve (Uttar Pradesh).	Protected Area.	110–185 masl.	Three major rivers susceptible to flooding, seasonal streams.	Suheli and Mohana rivers are moderately polluted with sewer discharge, detergents and fertilizers [177,178].	After a channel shift due to a flood in 2010, sandy open banks of the river became covered in woody vegetation, limiting nesting spot availability for both muggers and gharials [179]. To remedy this transition, in 2020, a project meant to build additional sand banks was carried out [179]. Muggers and gharials alike swiftly adopted these sand banks, although the authors of the project warn that this solution is only temporary [179].	The three major Protected Areas forming Dudhwa Tiger Reserve are separated by privately owned land and 125 villages within a 5 km boundary (as of 2001), further causing encroachment into the forests and difficulties in maintaining the reserve’s role as an ecological corridor [180].		Moderate.
Gir National Park (Uttar Pradesh).	Protected Area.	N/A	Seven major perennial rivers, reservoirs, smaller rivers and streams and 388 artificial water points [181].	N/A	N/A	There is conflict between the park and local communities, as Maldharis settled in Gir forest caused major damage to the park by overgrazing livestock. Eventually, they were relocated outside of the Protected Area in 1972, but encroachment and overgrazing remain issues [181].	Mugger crocodiles inhabit major reservoirs in Gir forest and several rivers [181].	Data deficient.

N/A—data necessary for assessment not available.

**Table 6 animals-14-00691-t006:** Muggers reintroduced in India.

Location	No of Individuals	State	Breeding Centre of Origin
Coringa Wildlife Sanctuary	3	Andhra Pradesh	- *
Nagarjunsagar-Srisailam Tiger Reserve	136	Andhra Pradesh	-
Manjira Wildlife Sanctuary	212	Telangana	Nehru Zoological Park, Manjira Wildlife Sanctuary Crocodile Breeding Center
Pakhal Wildlife Sanctuary	15	Telangana	-
Kinnersani Wildlife Sanctuary	33	Telangana	-
Kaweri South Wildlife Sanctuary	130	Tamil Nadu	Madras Crocodile Bank Trust
Hoggenakal	48	Tamil Nadu	-
Mundanthurai Wildlife Sanctuary	25	Tamil Nadu	-
Shivpuri National Park	25	Madhya Pradesh	-
Neyyar Wildlife Sanctuary	36	Kerala	
Similipal Tiger Reserve	390	Odisha	Nandanakanan, Ramatirtha
Mahanadi	112	Odisha	Nandanakanan, Ramatirtha
Gir National Park	857	Gujarat	Sasan, Gandhinagar

* Origin unknown.

**Table 7 animals-14-00691-t007:** Country summaries regarding mugger population status.

Country	Change from Historic Range	Population Size Trend	Habitat Suitability	Status of Captive Populations	Major Threats	Chance of Survival	Recommendations
Sri Lanka	Insufficient data	Stable	Moderate	No captive populations	1. Habitat degradation (land conversion, draining of wetlands). 2. Human–crocodile conflict (poaching, “preventative” killings, deaths in fishnets).	Moderate	1. Monitoring current range and status of muggers in Sri Lanka. 2. Creation of an intense education program on safe coexistence practices and muggers’ role in the ecosystem. 3. Identifying migration corridors and maintaining their permeability.
Bangladesh	Extinct in the wild	Extinct in the wild	Bad	29 muggers (lack of breeding success)	1. Lack of breeding success in the captive population. 2. High possibility of developing human–crocodile conflict after reintroduction.	Low	1. Identifying causes of lack of breeding success in the captive population, including genetic testing. 2. In case of future reintroduction, launching an educational program on safe coexistence prior to reintroduction.
Iran	No change	Stable	Moderate	Approximately 120 muggers (as of 2018) in three centers.	1. Climate change (droughts, floods). 2. Habitat degradation (water pollution).	Moderate	1. Prioritizing water security and combating climate change. 2. Maintaining captive crocodiles for gene preservation. 3. Monitoring water quality in known mugger habitats.
Pakistan	Insufficient data	Declining	Bad	Approximately 140 non-hatchlings (overstocked)	1. Climate change (drought, flooding). 2. Habitat degradation (water shortage, water pollution). 3. Human–crocodile conflict (“preventative” killings, killings for perceived competition, killings for sport, deaths in fishnets).	Low	1. Prioritizing water security and combating climate change. 2. Monitoring current range and status of muggers in Pakistan. 3. Creation of an intense education program on safe coexistence practices and mugger’s role in the ecosystem.
Nepal	Disappeared from Gaidahawa Tal, Jagadishpur Reservoir and Reu river system	Rising	Moderate	Breeding captive population in Gharial Breeding Center in Kasaran	1. Habitat degradation (siltation, eutrophication, invasive plant species, dam and barrage construction). 2. Climate change (flooding). 3. Human–crocodile conflict “preventative” killings, killings for perceived competition, deaths in fishnets).	Moderate	1. Monitoring current range and status of mugger in Nepal. 2. Egg collection and hatchling rearing to reduce loss of breeding success to flooding. 3. Creating an education program with emphasis on integrating local communities into crocodile conservation.
India	Insufficient data (likely shrinking)	Rising	Moderate	1968 individuals held in nine captive centers (as of 2021).	1. Habitat degradation (encroachment, water pollution, dam and barrage construction. 2. Climate change (water shortage). 3. Human–crocodile conflict (deaths in fishnets, opposition from local communities to conservation efforts, lack of safe access to water in rural areas).	High	1. Prioritizing water safety by better management of sewage and industrial discharge. 2. Creating an education program with emphasis on safe coexistence practices and integrating local communities into crocodile conservation.

## Data Availability

Data sharing not applicable.

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
