# Peer review of "Current State of Mugger Populations"

_animals, 2024, doi:10.3390/ani14050691_

Round 1

Reviewer 1 Report

Comments and Suggestions for Authors

The stated goal of this paper is to comprehensively determine mugger crocodile’s current range and assess risks. The authors summarize a wealth of information for each country within the mugger’s range and provide descriptions of threats and offer a prognosis for the future.  In general, the article is easy to follow and contains a lot of interesting and relevant information. I think there are opportunities to improve the impact by providing additional clarification on methods, terminology, and synthesis.  If there is someway to organize the summary information in a table by country, I think that would help to show at the big picture level in which countries muggers are at most risk (see end of comments for idea).  It would also be helpful if there was a supplementary table with the information summarized by protected area, but that could be a bit of work and is not necessary since the information is in the text.  The advantage of such a table would be the ability for readers to do a quick skim of the information.  At the end of the introduction (line 51) it states that this paper will determine status and survival changes, I would like to see tie back to this in the conclusion with relative ranking of survival chances (see table at end of comments). Basically, provide a summary to support this statement: “Based on our findings, chances of mugger crocodile’s survival vary greatly across its range” and end with specific recommendations.

Below are additional comments/suggestions.

Line     Comment

34        What about including an overall figure with the historic range/distribution of muggers for big picture context? This would help orient the reader overall.

52        Typo- threaths should be threats.

53        Prognosis (a forecast on the likely outcome) might be a better word than prediction (a think predicted a forecast).

53/54   Need a section describing the methods of the literature review in the body of the manuscript.  Suggest including what sources were used (google scholar, web of science, personal collections, CSG files, etc.), what key words were used, what years were included.  This is relevant as there seem to be some potentially useful references from 2022/2023 that I saw in a quick google scholar search that are not cited.

54        My note on my first read was I wanted to see a table with how each of the categories was characterized to come up with an overall Valorization score for example for status something that quantified number or percent of area in each of the categories. When I got to the country sections I realized that this was a more qualitative description of the habitats in the country- not so much what I would call valorization.  Suggest changing the title of this section to habitat characterization and suitability.

54        Suggest that section 2 is Materials & Methods as seems to be required.  Then within that start with the literature review followed by a section that describes how the information is presented for each country.

Animals now accepts free format submission:

  • We do not have strict formatting requirements, but all manuscripts must contain the required sections: Author Information, Abstract, Keywords, Introduction, Materials & Methods, Results, Conclusions, Figures and Tables with Captions, Funding Information, Author Contributions, Conflict of Interest and other Ethics Statements. 

Something like this:

2. Materials & Methods

We conducted extensive literature review by…. The information was then synthesized for each of the six countries within the mugger’s historic range and the information presented following the format co-specifics, historical range, current range, habitat description and quality (habitat valorization), threats, captive populations, other conservation efforts, prospectus and recommendations (predictions).

Then describe what is meant by the headings listed for each country for example, historical and current range.  It seems in the text there is a mixing of range and status.  It is also not clear what the cutoff for historic is compared to current and sometimes information from similar time periods is presented in both sections.  I think it would improve the presentation to put the information in the same timeframe context and to separate out range from population size/status.  Perhaps the sections are relabeled range and status.  To me range speaks to geographic extent while status is more about population size in whatever geographic extent is of interest.  The description of what is included in the habitat valorization methodology would be included here.

Section 3 would then be country reports or country summaries with each of the headings as described above and used already but as a second level heading.  3.1.1 for example for Sri Lanka historical range/status and 3.2.1 for Bangladesh historical range/status.

Question on the order of the countries- is there any significance of the order?  Geographic? Population size? Vulnerability?  If so, include that in the intro where you describe summarizing the information by country.

137      This applies to all maps.  They would benefit from more information in the caption.  Could you include boundaries of the region/biogeographic zones mentioned in the text to give the reader an idea of how much of the country actually has habitats that could be suitable at those broader ecological regions?  This would be helpful for readers not a familiar with the area. Check that key places mentioned in the text are on the maps with the same names.  It seems that what is shown on the maps are where muggers are known to occur now? (current range).  Clarify what is meant by habitat on the maps.

General comment- When possible be specific rather than using numerous or several.  Suggest doing a search for vague words like that and where possible replace with specifics.

General comment- Provide a little more location information for readers not familiar with the region.  For example for India reference the location Kaweri River to the appropriate map/state (Tamil Nadu?).

General comment- Check for where clarification might be needed in terminology.  For example, is it historic range or range and population size?  Are animals relocated or translocated (there are different).

One suggested presentation of summary information

Country

Change from historic range

Major Threats

Status of Captive populations

Prognosis summary

Chance of survival

Recommendations

Sri Lanka

Decline by xx%

Bangladesh

Decline by 100%

Iran

No change

Pakistan

Nepal

India

Reviewer 2 Report

Comments and Suggestions for Authors

Comments to the authors:

In the manuscript “Current state of the Mugger crocodile in the world”, the authors provide a very thorough and detailed literature review of the current distribution, habitat, conservation efforts, and threats of Mugger crocodiles (Crocodylus palustris) throughout the world.

Firstly, I would like to commend the authors, as it is obvious that a considerable amount of time and effort has been put into the creation of this manuscript. Overall, I found that the manuscript was well written and that it provides an important contribution to the conservation of Muggers and crocodylians in general. While I did encounter some minor grammatical and spelling errors throughout the manuscript, with how large the manuscript currently is, such errors should be expected. I have provided comments below for wherever I have detected a minor error. However, I would still recommend that the authors give the manuscript a thorough read-through for any I may have missed.

A major comment I have for the manuscript is that it is not clear what is the purpose of the habitat valorization and why it was conducted. Currently, beyond providing a structure for the authors to discuss where Muggers are found within each country, it is unclear what other benefits habitat valorization has provided over a traditional literature review. Does it help rank habitat suitability between countries? Or is it a specific way to make knowledge of Mugger habitat suitable for the relevant stakeholders?

Further, while the manuscript is based on quite an extensive literature review, the authors do not provide any details about how this was conducted (i.e., databases and keywords used). The inclusion of this information would be beneficial not only to help validate the results of the findings but also to provide a guide for future work interested in replicating this approach on other species.

I also have concerns about the current length of the manuscript. At 32 formatted pages, it is currently quite the epic to read, which I feel is, unfortunately, to the detriment of the manuscript. While I commend the authors for their thoroughness, there is simply too much information being given to the reader at once, making it difficult to identify key aspects and follow the manuscript. Where possible, I recommend that the authors condense the text, particularly in the results of the habitat valorization of each country.

Specific comments:

Line 24: Should this be eight instead of seven? For values under ten, it is also typically recommended that they are spelt out in sentences and not just a numeral.

Line 72-73: Please include a citation for the reports of Muggers not being found above 420 masl.

Line 73-76: I found this sentence awkward and difficult to read. I would recommend that the authors restructure it to first introduce that muggers have been observed at relatively high altitudes in the past, which then suggests that a lack of habitat has prevented this from being a more common occurrence.

Line 119: Should this instead be “human killing”?

Line 146: “C. palustris” should be italicised.

Line 151-152: This is a bit of an awkward sentence. Perhaps update it to “At present, muggers are mostly confined to the first peneplain of Sir Lanka (Figure 1) in the dry zone below 100 masl”.

Line 356: Update to “are”.

Line 371: Update to “unstable”.

Line 528: Should this be “poaching”?

Line 647: “By” is misspelled.

Line 704: “The” missing between creating and Hub.

Line 734-738: I recommend either adding “A” before “study” or changing the start of the sentence to “Chang et al. (2013) attempted to…”. I would also recommend changing “detected” to “detecting” and including an “and” before indicating.

Line 875: Change to “three” instead of "3”. A number at the start of a sentence should always be spelled out rather than noted as a numeral.

Line 878: “Annually” is misspelt.

Line 879-880: “Of faster growth than native species” is not necessary at the end of this sentence.

Line 911-913: Update to “Locals depend on the reserve for firewood, grasses, timber and fish, with thousands entering Koshi Tappu WR daily, leading to significant human-wildlife conflict”.

Line 939-941: I feel like this sentence is missing something between “opened up” and “densely inhabited”.

Line 959-960: I found this sentence a little clunky to read. Perhaps consider rephrasing it to “According to the Climate Risk Index, Nepal ranks as the 10th most affected country in the world by climate change, particularly by flooding events”.

Line 1032: Replace “36” with “Thirty-six” as I indicated above.

Line 1044-1046: I recommend replacing with “Between 2013 and 2015, muggers were recorded at 27 locations within Kheda and Anand, with an overall density of 14.31 individuals per 100km2 [134]. Of these observations, 29% were recorded within the village Deva [134].”

Line 1055: I recommend adding “A” before survey.

Line 117: “Length” is misspelt.

Line 1400-1401: I recommend adding “a” between caused and rise.

Line 1439: Remove only.

Line 1455: Not should be before been.

Line 1498: “Lengthy” is misspelled.   

An overarching comment I have for the manuscript figures is that, if possible, it would be beneficial to include a visual representation of the current distribution of muggers within each country, as used in most field guides. It is not initially clear that the figures are currently representing the known distribution of muggers, specifically for countries like Sri Lanka.

A north arrow and scale bar, which are key cartographic elements, are not currently included in any figure.

Finally, I also recommend varying the colour of your symbology, as it is currently difficult to differentiate each habitat type due to only varying shades of blue being used.

Comments on the Quality of English Language

Overall, I found that the manuscript was well written and that it provides an important contribution to the conservation of Muggers and crocodylians in general. While I did encounter some minor grammatical and spelling errors throughout the manuscript, with how large the manuscript currently is, such errors should be expected. I have provided comments below for wherever I have detected a minor error. However, I would still recommend that the authors give the manuscript a thorough read-through for any I may have missed.

Reviewer 3 Report

Comments and Suggestions for Authors

The manuscript Current State of the Mugger Crocodile in the World, which contains information on records of the mugger crocodile in the areas where it is distributed, is a compilation of this published information. Because this species of crocodile is seriously threatened by several factors, among which the most important are habitat loss and illegal hunting, I believe the compilation that the authors made to organize the information and present it according to the site is interesting. It seems to me a significant effort by the authors to organize all the information collected in this regard. However, the manuscript needs a much more rigorous and systematic analysis of the information, this would help provide relevant information that contributes to the conservation of the species. As the manuscript stands, it is only an organized presentation of the information and does not contribute anything to the main theme of the title. Below I briefly explain my observations.

Title

According to the information contained in the manuscript, I suggest changing the title to conservation status or the current state of crocodile populations. Ideally, authors should think of a title that more appropriately reflects the content of their study.

Introduction

The introduction should be more complete, the authors mention the generalities of the aspects that have reduced and isolated crocodile populations, which is fine, but that information can be applied to some other species as well. I suggest that the authors also briefly mention why the habitat is important, for example, during the nesting season crocodiles need a certain type of substrate to make their nests in addition to nearby bodies of water. This is the most vulnerable and most important stage in the life cycle because if the plants, humidity, and conditions that favor nest production as well as hatching success disappear, crocodile populations will face probable extinction. Then it information can be understood that the characteristics of the habitat are important. The authors must highlight different habitat characteristics that are crucial for the survival of Mugger crocodile populations.

Materials and methods

This section is missing in the manuscript, since this section explains clearly and in detail how the authors conducted their information searches: what were the academic search engines used, which were the keywords used, which were the search criteria of inclusion and exclusion used, etc.

I suggested conducting an appropriate analysis. It may be in the form of a meta-analysis, according to the objective of the review of the existing information. In other words, I suggested that the manuscript provides more than just a summary of the information obtained from the literature.

In the “Habitat valorization” section, I would expect that a value would be assigned, either qualitative or quantitative, according to previously defined criteria. This would help to understand the importance of the different components of the habitat that are important for crocodiles. In the manuscript, the authors only describe the habitat where Mugger crocodile populations or individuals have been recorded, they do not assign importance values.

Maps

The maps must serve to locate the reader geographically. I recommended that the authors incorporate all the necessary elements on a map, with the aim that the reader can locate on the map where each site is.

In general, the manuscript needs a review of language as well as format according to the author's guide for the journal. Below I detail some errors found in the text.

Sri Lanka

It is necessary to include geographic information about the site, or at least what country in which it is located. The map must contain the corresponding legends that help better geographical understanding (coordinates, scale, etc.)

Line 146

I suggested starting the sentence with a word that is not the abbreviated genus of the species. Scientific names must be written in italics.

Line 148

The format of the citation is not the one specified by the journal.

Line 155

I suggest unifying the common name as mugger throughout the text.

Line 156

The format of the citation is not the one specified by the journal.

Line 158

The format of the citation is not the one specified by the journal.

Line 163

What do the authors mean by the term “notable populations”?

Lines 164 and 165

I suggest deleting this paragraph. It has no connection with the previous information. It is more of an a priori judgment of the authors.

Line 219

The format of the citation is not the one specified by the journal.

Line 226

Scientific names must be written in italics.

Line 229

The format of the citation is not the one specified by the journal.

Line 233

It should say  million.

Line 238

The format of the citation is not the one specified by the journal.

Line 243

The format of the citation is not the one specified by the journal.

Bangladesh

It is necessary to include geographic information about the site, or at least what country in which it is located. The map must contain the corresponding legends that help better geographical understanding (coordinates, scale, etc.)

Line 401

The format of the citation is not the one specified by the journal.

Line 402

The format of the citation is not the one specified by the journal.

Line 404

The format of the citation is not the one specified by the journal.

Line 405

The format of the citation is not the one specified by the journal.

Line 408

The format of the citation is not the one specified by the journal.

Line 412

The format of the citation is not the one specified by the journal.

Line 413

The format of the citation is not the one specified by the journal.

Line 418

The format of the citation is not the one specified by the journal.

Iran

It is necessary to include geographic information about the site, or at least what country in which it is located. The map must contain the corresponding legends that help better geographical understanding (coordinates, scale, etc.)

Pakistan

It is necessary to include geographic information about the site, or at least what country in which it is located. The map must contain the corresponding legends that help better geographical understanding (coordinates, scale, etc.)

Line 610

The format of the citation is not the one specified by the journal.

Line 618

The format of the citation is not the one specified by the journal.

Line 628

The format of the citation is not the one specified by the journal.

Line 633

Do not start a sentence with a number.

The format of the citation is not the one specified by the journal.

Line 636

The format of the citation is not the one specified by the journal.

Line 691

The format of the citation is not the one specified by the journal.

Line 720

The format of the citation is not the one specified by the journal.

Line 754

The format of the citation is not the one specified by the journal.

Line 770

The format of the citation is not the one specified by the journal.

Line 772

Review the guide for authors regarding writing numbers less than nine.

Line 773

The format of the citation is not the one specified by the journal.

Line 777

The format of the citation is not the one specified by the journal.

Line 788

Review the guide for authors regarding writing numbers less than nine.

Line 794

Review the guide for authors regarding writing numbers less than nine.

Line 796

Review the guide for authors regarding writing numbers less than nine.

Line 799

The format of the citation is not the one specified by the journal.

Nepal

It is necessary to include geographic information about the site, or at least what country in which it is located. The map must contain the corresponding legends that help better geographical understanding (coordinates, scale, etc.)

Line 825

Review the guide for authors regarding writing numbers less than nine.

Line 836

The format of the citation is not the one specified by the journal.

Line 842

The format of the citation is not the one specified by the journal.

Line 843

The format of the citation is not the one specified by the journal.

Line 880

The format of the citation is not the one specified by the journal.

Line 934

The format of the citation is not the one specified by the journal.

India

It is necessary to include geographic information about the site, or at least what country in which it is located. The map must contain the corresponding legends that help better geographical understanding (coordinates, scale, etc.)

Line 996

Review the guide for authors regarding writing numbers less than nine.

Line 1028

The format of the citation is not the one specified by the journal.

Line 1042

The format of the citation is not the one specified by the journal.

Line 1135

Review the guide for authors regarding writing numbers less than nine.

Line 1156

I recommended not to start a sentence with a number.

Reviewer 4 Report

Comments and Suggestions for Authors

The authors are to be commended for assembling the extensive information, both historical and current, on Mugger crocodiles. The paper aims to assess population status, threats and “habitat valorization”. However, I am concerned that it resembles more of a collection (review) of available information, rather than a strong emphasis on a clear assessment of the available information. 

General comments include: 

  1. With regard to population status, the paper largely duplicates what the IUCN Red List assessments have already done for Muggers. That is, past and current population trends (and estimates for future) are used to assess the risk of extinction in the future. That status varies between countries and localities is already well established for Muggers (as it is for many other crocodilians).  

  1. I am not sure that I agree that Iran should be considered a stronghold for the species (as stated in Conclusions), given that this population is at the westernmost extent of the species’ range, and habitats are very harsh, and likely to impact future population status. Information from Nepal suggests an increasing Mugger population, particularly in protected areas, and may be significant for the species’ long-term conservation.  

  1. Red List assessments also take into account range/distribution and potential impacts of habitat loss, etc., which are dealt with in the paper as well.  

  1. Attention needs to be given to the maps. For example, the map of Bangladesh purportedly shows major habitats for the species – but it is considered extinct in that country. It is impossible to relate what is in the text to what the map is supposed to show. 

  1. In various places, “threats” identified decades ago are cited as if they are still operating now. For example, “The drastic decline of mugger crocodile numbers has been attributed to high demand for their skin and meat. According to Whitaker and Whitaker (1984), fishermen in Sri Lanka also kill crocodiles for meat, sometimes as many as 20 in a day”. This citation is 39 years old! Is the situation still the same now? 

  1. Some information provided appears somewhat superfluous as it is very much out of date. The information on numbers of Muggers in Nepalese and Bangladeshi zoos, from 1994 and 2009 respectively, is outdated, and does not necessarily add anything to the assessment. If required, at least it can be prefaced by something like “Current information of captive stocks in …… are unknown, however they were …….. in ……..”. 

  1. The first part of opening sentence of the Simple Summary and lines 46-48 are not correct, and therefore misleading – “Mugger crocodile is an ecologically important keystone species …..”. “Meanwhile, mugger crocodiles perform a role in maintaining the structure and function of freshwater ecosystems as top predators and keystone species affecting all the animals below them in the food chain”. Somaweera et al. postulated otherwise. There is no evidence that crocodilians perform such a role. Indeed, in many cases they were reduced to minor proportions of their historical abundance, and there was no collapse of ecosystems, etc. 

  1. Some population estimates may need to be checked. For example, the India population is quoted as being around 4000 non-hatchlings (lines 1013-1014). This may in fact be the estimated number of adult Muggers. 

Round 2

Reviewer 3 Report

Comments and Suggestions for Authors

After reviewing the manuscript, I consider that the authors corrected the formatting errors and added the suggested information, however, there is an important issue that I mentioned in my previous review which is that I consider that the manuscript is a set of information without a rigorous analysis of this to help understand how this information compiled from other studies can help establish the status of crocodile populations.

In that case, I suggest that the manuscript be suitable as a review article, and not as original research.

Author Response

We would like to first once again express our gratitude to the reviewer for their input into improving the manuscript. 

We agree with the reviewer's comment and would like to inform that we have indeed sumbitted the manuscript as a review article.

Reviewer 4 Report

Comments and Suggestions for Authors

The manuscript is greatly improved. Some minor editing of the English is required.

Some specific comments include:

Line 132: insert “m asl”

Line 395: Elsewhere “Bangladesh”, suddenly “People’s Republic of Bangladesh” appears

Line 390 states reintroduction took place in “2005”, but at Line 478 it is “2004”

Line 561 “Islamic Republic of Iran” but “Iran” elsewhere

Line 743: “Islamic Republic of Pakistan” but elsewhere “Pakistan”

Line 866: “While Pakistan is a CITES party …” – it is irrelevant that Pakistan is a Party to CITES – the activity is illegal relative to Pakistan and/or Iranian law.

Line 971: “Federal Democratic Republic of Nepal” but elsewhere “Nepal”

Line 1651 – check area – “1 736 2952ha”

Comments on the Quality of English Language

The manuscript is greatly improved. Some minor editing of the English is required. Some specific comments include:

Line 132: insert “m asl”

Line 395: Elsewhere “Bangladesh”, suddenly “People’s Republic of Bangladesh” appears

Line 390 states reintroduction took place in “2005”, but at Line 478 it is “2004”

Line 561 “Islamic Republic of Iran” but “Iran” elsewhere

Line 743: “Islamic Republic of Pakistan” but elsewhere “Pakistan”

Line 866: “While Pakistan is a CITES party …” – it is irrelevant that Pakistan is a Party to CITES – the activity is illegal relative to Pakistan and/or Iranian law.

Line 971: “Federal Democratic Republic of Nepal” but elsewhere “Nepal”

Line 1651 – check area – “1 736 2952ha”
